# Biophysical Properties of Somatic Cancer Mutations in the S4 Transmembrane Segment of the Human Voltage-Gated Proton Channel hH_V_1

**DOI:** 10.3390/biom15020156

**Published:** 2025-01-21

**Authors:** Christophe Jardin, Christian Derst, Arne Franzen, Iryna Mahorivska, Thomas E. DeCoursey, Boris Musset, Gustavo Chaves

**Affiliations:** 1Center of Physiology, Pathophysiology and Biophysics-Nuremberg, Paracelsus Medical University, 90419 Nuremberg, Germany; christophe.jardin@pmu.ac.at (C.J.); christian.derst@pmu.ac.at (C.D.); iryna.mahorivska@klinikum-nuernberg.de (I.M.); boris.musset@pmu.ac.at (B.M.); 2Institut für Biologische Informationsprozesse, Molekular-und Zellphysiologie (IBI-1), Forschungszentrum Jülich, 52428 Jülich, Germany; a.franzen@fz-juelich.de; 3Department of Physiology & Biophysics, Rush University, Chicago, IL 60612, USA; thomas_decoursey@rush.edu

**Keywords:** voltage-gated proton channels, hH_V_1, somatic mutations, cancer, tumor cells, pH-homeostasis, S4 transmembrane domain, voltage-sensing arginines

## Abstract

Somatic mutations are common in cancer, with only a few driving the progression of the disease, while most are silent passengers. Some mutations may hinder or even reverse cancer progression. The voltage-gated proton channel (H_V_1) plays a key role in cellular pH homeostasis and shows increased expression in several malignancies. Inhibiting H_V_1 in cancer cells reduces invasion, migration, proton extrusion, and pH recovery, impacting tumor progression. Focusing on *HVCN1*, the gene coding for the human voltage-gated proton channel (hH_V_1), 197 mutations were identified from three databases: 134 missense mutations, 51 sense mutations, and 12 introducing stop codons. These mutations cluster in two hotspots: the central region of the N-terminus and the region coding for the S4 transmembrane domain, which contains the channel’s voltage sensor. Five somatic mutations within the S4 segment (R205W, R208W, R208Q, G215E, and G215R) were selected for electrophysiological analysis and MD simulations. The findings reveal that while all mutants remain proton-selective, they all exhibit reduced effective charge displacement and proton conduction. The mutations differentially affect hH_V_1 kinetics, with the most pronounced effects observed in the two Arg-to-Trp substitutions. Mutation of the first voltage-sensing arginine (R1) to tryptophan (R205W) causes proton leakage in the closed state, accelerates channel activation, and diminishes the voltage dependence of gating. Except for R205W, the mutations promote the deactivated channel configuration. Altogether, these data are consistent with impairment of hH_V_1 function by mutations in the S4 transmembrane segment, potentially affecting pH homeostasis of tumor cells.

## 1. Introduction

Somatic mutations are random mutations occurring in somatic cells throughout a person’s lifespan [1]. They may arise from unrepaired replication errors or from external factors, such as UV light or chemicals. In tumor cells, up to 20,000 somatic point mutations per cell may significantly influence the respective cancer genome [2,3]. Somatic mutations in cancer cells are usually divided into “driver mutations”, which induce cell proliferation and tumor growth, and “passenger mutations” without significant influence on tumor growth [4]. In recent years, a more detailed view on passenger mutations showed that at least some of them may function as “mini-drivers”, providing a small selective advantage to cancer cells without being necessary for tumor growth [5]. On the other hand, other passenger mutations may have deleterious effects on tumor progression, especially when many of them accumulate within the cancer genome [6]. Therefore, at least four categories of somatic cancer mutations (“drivers”, “mini-drivers”, “silent passengers”, and “deleterious passengers”) can be distinguished.

Over the past decades, evidence has accumulated that ion channels can act as tumor drivers in several cancer types [7]. The protumorigenic effects of ion channels include cell cycle regulation, cell growth regulation, control of the intracellular Ca^2+^ level, tumor cell movement, metastasis, and apoptosis [8,9,10,11,12,13,14]. In this context, the voltage-gated proton channel has also been a target for potential pharmacological applications in cancer therapy [15,16].

The microenvironment of the tumor strongly depends on the balance of proton uptake and proton extrusion. Therefore, control and homoeostasis of the intracellular pH is crucial. A hallmark of many tumor cells is an increased glucose uptake with a subsequent fermentation to lactate as an additional energy source, thereby producing a substantial quantity of cytosolic protons, known as the Warburg Effect [17,18,19]. To overcome the cytosolic acid stress, the expression of several proton transporters and exchangers is upregulated [20,21,22]. These transport mechanisms enhance acid extrusion to restore intracellular pH to its physiological value. However, this balance is delicate and intracellular pH becomes often more alkaline [23]. Furthermore, intracellular pH may exert some control over the cell cycle [24] and the change in the pH inside or outside a cancer cell potentiates their development into more malignant subtypes. The growth of tumors is further supported by acidic extracellular pH which helps cancer cells to evade immune surveillance and affects T-cells, natural killer cells, and macrophage function [25]. As a consequence, the voltage-gated proton channel (H_V_1) has become a focus of research, as several reports indicate that it plays a key role in acid/proton extrusion in certain tumors, such as leukemia, lymphoma, glioma, colorectal cancer, and breast cancer [26,27,28,29]. Moreover, H_V_1 may even function as a useful biomarker in diagnosis and prognosis for some cancer types, such as colorectal cancer [30] and lymphoma [31].

Focusing on somatic cancer mutations of *HVCN1*, the gene of the human voltage-gated proton channel (hH_V_1), we analyzed five selected somatic mutations within the S4 segment to evaluate their possible influence on the cancer cell phenotype. The S4 transmembrane helix contains the channel’s putative voltage-sensor motif (VSM), proton pathway, and the adjacent C-terminal part of hH_V_1. The five mutations in the S4 alpha helix, three of them located in the VSM (R205W, R208W, R208Q) and two C-terminally from it (G215E and G215R), were characterized using patch-clamp and molecular dynamics. The five variants were found to alter channel function, with R205W and R208W showing pronounced changes in gating while maintaining proton selectivity. These mutants alter hH_V_1 channel function in several manners, with a general result comparable to hH_V_1 inhibition. Based on a hypothetical correlation between hH_V_1 activity and cancer progression, we predict that these mutations might affect tumor cells by affecting pH homeostasis.

## 2. Methods

### 2.1. Database Analysis

Three databases (COSMIC, TCGA, and ICGC) where screened for somatic cancer mutations of the *HVCN1* gene [32,33,34]; see Appendix A. Three different sets of mutations were collected from each database: missense mutations, sense mutations, and mutations introducing a stop codon. Larger deletions or insertions, splice site mutations, and variations in *HVCN1* copy number were not analyzed in this study. To avoid duplicate database entries, mutations found in the TCGA and ICGC databases were only included in our dataset as a new mutation when they were not found in the COSMIC database.

To visualize the distribution of somatic cancer mutations within the *HVCN1* gene and to identify possible mutational “hot spots”, for each residue, the number of mutations at a given residue and the surrounding three residues upstream and downstream were calculated (“7-residue window”).

### 2.2. S4 Multiple-Sequence Alignment

A multiple-sequence alignment (MSA) of the S4 transmembrane segment from several proton channels and other voltage-sensing proteins was performed using CLUSTAL OMEGA (v1.2.4) (https://www.ebi.ac.uk/jdispatcher/msa/clustalo (accessed on 9 September 2024)). The alignment was constructed with the FASTA sequences in the UniProtKB/Swiss-Prot databases: *Homo sapiens* hH_V_1 (Accession: Q96D96), *Mus musculus* mH_V_1 (Accession: NP_001035954), *Nicoletia phytophila* NpH_V_1 (Accession: KT780722.1), *Ciona intestinalis* CiH_V_1 (Accession: NP_001071937), *Aplysia californica* AcH_V_1 (Accession: XM_005100609), *Aplysia californica* AcH_V_2 (Accession: XM_005093050), *Aplysia californica* AcH_V_3 (Accession: XM_005094218), *Crassostrea gigas* H_V_4 (Accession: XM_011429833), *Ciona intestinalis* voltage-sensing phosphatase (Accession: NP_001028998), *Drosophila melanogaster* Shaker (Accession: CAA29917), and *Homo sapiens* voltage-gated sodium channel 1.1 (Accession: P35498).

### 2.3. Computational Methods

The wild-type (WT) and mutant hH_V_1 channels were investigated with computational methods both in deactivated and in activated configurations. The channel might have more than one closed and one open conformation. Here, we use the terms “deactivated” and “activated” to distinguish configurations that we assume to be able to conduct protons across the membrane (activated or open) or not (deactivated or closed). An essential structural feature that distinguishes the two configurations is the outward displacement of helix S4 during activation. Although there are still controversies about the extent of S4’s displacement, at least the second (R2) of the three (R1–R3) voltage-sensing arginines in S4 pass across the hydrophobic gasket (HG) that separates the inner and outer vestibules of the channel. The HG is formed by four hydrophobic residues: V109 in S1, F150 in S2 and V178, and V179 in S3 [35]. The displacement of R2 from the internal to the outer external aqueous vestibule during gating is supported by several studies [36,37,38,39].

#### 2.3.1. Construction of Structural Models

To date, there is little experimental information on the structure of the hH_V_1 channel. Therefore, we generated structural models of a deactivated and an activated configuration. Details about the generation of these models and the introduction of the mutations are given in the Appendix A.

#### 2.3.2. Preparation of the Systems for Molecular Dynamics Simulations

The systems for the molecular dynamics (MD) simulations of the WT and mutants in both configurations were prepared following the approach used in previous works [36,40] (see Appendix A).

#### 2.3.3. Molecular Dynamics Simulations

Each system was simulated with MD for 5 µs without constraints. The coordinates of the atoms were recorded at regular time intervals for analysis. We will use the terms snapshot or frame interchangeably to refer to one record. For details about the simulation and analysis, see Appendix A.

### 2.4. Heterologous Expression

The hH_V_1 cDNA was synthesized and inserted into a pEX-A2 plasmid (Eurofins/Genomics, Ebersberg, Germany). By the use of *Bam*HI and *Eco*RI restriction sites at the 5′ and 3′ ends, the gene was subsequently subcloned into the pQBI25-fC3 vector and a GFP tag was added to the N-terminal region of the construct (WT hH_V_1). Site-directed mutagenesis by PCR overlapping was performed afterwards to generate each of the mutant clones. Each clone was sequenced commercially to confirm mutations. HEK293 and tSA201 cell lines were transfected when they reached approximately 85% confluency using 1 μg of cDNA and polyethylenimine as the transfection reagent (Sigma, St. Louis, MO, USA). The cells were incubated for 12 h at 37 °C with 5% CO_2_, after which they were trypsinized and re-plated onto glass coverslips at a low density. Patch-clamp recordings were conducted on the same day or the next day, selecting single green fluorescent cells for measurement.

### 2.5. Electrophysiology

Patch-clamp recordings followed established protocols. The recordings were performed using an EPC 10 patch-clamp amplifier (HEKA Elektronik, Lambrecht, Germany). Borosilicate glass capillaries (GC 150TF-10; Harvard Apparatus, Holliston, MA, USA) were used to fabricate the patch pipettes with a Flaming Brown automatic pipette puller P-1000 (Sutter Instruments, Novato, CA, USA). The pipette tips were heat-polished to achieve resistances between 5 and 9 MΩ when in contact with the solutions. Electrical contact was established using a chlorinated silver wire, with an agar bridge (made with Ringer’s solution) connecting the pipette solution to the bath. Ringer’s solution (160 mM NaCl, 4.5 mM KCl, 2 mM CaCl_2_, 1 mM MgCl_2_, 5 mM Hepes, pH 7.4) was used in the bath to form seals, and the potential was set to zero once the pipette was positioned above the cell. The solutions used for whole-cell recordings, both in the pipette and the bath, contained 100 mM of a buffer near its pK_a_ with tetramethylammonium (TMA^+^) (Merck KGaA, Darmstadt, Germany) and methanesulfonate (CH_3_SO_3_^−^) (Merck KGaA, Darmstadt, Germany) as the primary ions, 1 mM EGTA SERVA Electrophoresis GmbH, Heidelberg, Germany), and 1–2 mM Mg^2+^ (Merck KGaA, Darmstadt, Germany), with an osmolality of 300 mOsm·kg^−1^. Buffers included MES (Carl Roth GmbH, Karlsruhe, Germany) at pH 5.5 and 6.0, BIS-TRIS (Carl Roth GmbH, Karlsruhe, Germany) at pH 6.5, PIPES (Carl Roth GmbH, Karlsruhe, Germany) at pH 7.0, and HEPES (SERVA Electrophoresis GmbH, Heidelberg, Germany) at pH 7.5. Seal resistances normally exceeded 3 GΩ. Currents are presented without leak subtraction or correction for liquid junction potentials. Data acquisition occurred at temperatures between 19 °C and 22 °C. The threshold potentials (*V*_thres_) were determined from families of depolarizing pulses by identifying the voltage (*V*) at which the first tail current appeared upon membrane repolarization. Reversal potentials (*V*_rev_) were determined using two methods: directly by the zero current when *V*_thres_ was negative relative to *V*_rev_ and by the tail current method when *V*_thres_ was positive relative to *V*_rev_. Overexpressing the channels in small cells led to substantial proton currents, causing significant pH changes inside the cells due to proton depletion or proton accumulation [41]. Since proton channel gating is highly pH-sensitive, these sources of error must be minimized. To address this, pulse families of varying lengths were employed, using longer pulses for voltages near *V*_thres_ where activation was slow and shorter pulses at more positive voltages. In the case of channels that either activate at potentials more negative than *V*_rev_ or have very slow deactivation, the interval at the holding potential (*V*_hold_) between pulses was increased to ensure that the currents at the holding voltage (*I*_hold_) reached a steady state. Recordings were saved on hard drives.

### 2.6. Biophysical Characterization

Proton currents from transfected cells were analyzed using Origin software (Origin 2023, Northampton, MA, USA). Activation time constants (*τ_act_*) and maximal proton currents (*I*_max_) were obtained by fitting the currents to a rising exponential function:(1)$$I=A\cdot e^{(t/\tau )}+C$$
where *τ* is the time constant of activation, *I* is the current, and *A* and *C* are constants. In these fits, any initial delay was disregarded, and the remaining current generally followed a single exponential. The maximal proton conductance (*g*_H,max_) was calculated from the quotient of the extrapolated steady-state *I*_max_ (A) and the corresponding driving force (*V* − *V*_rev_) for each pH condition in each measured cell. *V*_rev_ was determined each time the solution was exchanged.

Deactivation time constants (*τ_deact_*) were obtained from tail currents recorded at voltages between −100 and 0 mV after short depolarizing pulses using Equation (1). In cases where the tails showed two components, only the slow component was used for further calculations.

Conductance density for each cell was determined as the quotient of *g*_H,max_ and the respective cell capacitance (*C_m_*).

Gating charges (*e*_0_) were obtained by the limiting slope method from *g*_H,max_-*V* curves at low conductive states, i.e., potentials close to *V*_thres_, as previously reported in proton channel analysis [42,43,44]. Low conductances were obtained either by applying depolarizing protocols with 2 mV/step increments from voltages close to *V*_thres_ or by applying the limiting slope to the most negative/steepest *g*_H_-*V* decade.

Proton conductance was normalized to *g*_H,max_ and fitted with a Boltzmann function of the following form:(2)$$\frac{g_{H}}{g_{H,max}}={\left[1+e^{\left(\frac{V_{0.5}-V}{k}\right)}\right]}^{-1}$$
where $\frac{g_{H}}{g_{H,max}}$ is the relative proton conductance, *V*_0.5_ is the half-activation potential, *V* is the membrane voltage or test voltage, and *k* is the slope factor.

The selectivity for protons was evaluated by comparing the measured *V*_rev_ with the Nernst potential for protons (*E*_H_) at the experimental ΔpH, in a pH range from 5.5 to 7.5. Under the different pH gradients tested (ΔpH = 1, 0.5, 0, and −1), the number of measurements per pH condition ranged from *n* = 3–7, except for the R205W, R208Q, and G215R mutants at ΔpH = 1, where *n* = 2.

All calculations for conductance density, gating charges, *V*_thres_, *V*_0.5_, and the kinetics of activation and deactivation were made under symmetrical pH conditions (ΔpH = pH_o_ − pH_i_ = 0). All values are presented as the arithmetic mean ± standard error of the mean (S.E.M.), with the exception of the *V*_rev_ vs. *E*_H_ plot, where values are shown as the arithmetic mean ± standard deviation (S.D.).

### 2.7. Kinetic Analysis

To compare the kinetics of activation and deactivation of the different mutants, membrane potentials were selected at which either the accumulation or depletion of protons is minimized. Subsequently, the analysis was standardized by selecting specific potentials of −40 mV and *V*_0.5_ at ΔpH = 0 to study deactivation and activation kinetics, respectively. In this way, the standardization of driving forces (|*V* − *V*_rev_|) was ensured for the closing process, and the evaluation of kinetics parameters at the same open probability was ensured for activation.

To quantify the effect of each mutation on the gating kinetics relative to the WT channel, perturbation energies were calculated as(3)$$∆∆G=-RT\ln\frac{\tau_{WT}}{\tau_{mut}}$$
where *R* is the universal gas constant, *T* is the absolute temperature, and *τ_WT_* and *τ_mut_* are the time constants of activation and deactivation for the wild-type and mutant channels at *V*_0.5_ and −40 mV, respectively. Values are shown as mean ± S.E.M.; *n* = 5–8.

The voltage dependence of the kinetics (*U*_τ_), expressed as mV per *e*-fold change, was determined from the slope of linear fittings of the kinetics plot (*τ*-*V*) within a range of −100 mV to +20 mV for deactivation kinetics and 0 mV to +100 mV for activation kinetics. Values are shown as mean ± S.E.M.; *n* = 4–6. In cases where linearity was lost, i.e., at potentials close to *V*_rev_ where signal resolution is compromised, only the voltage range showing linearity was considered for analysis.

The relationship between the voltage dependence of deactivation and activation was determined as the ratio (θ) between the slopes of both components:(4)$$\theta =\frac{U_{\tau ,deact}}{U_{\tau ,act}}$$
where *U_τ,deact_* and *U_τ,act_* are voltage-dependent kinetics of deactivation and activation, respectively. θ values are shown as mean ± S.E.M.; *n* = 5–7.

### 2.8. Evaluation of Proton Leak in the Deactivated State

For the different mutants, the proton leakage in the closed (deactivated) state was determined by exchanging the external buffer solution during test pulses to monitor changes in the holding current (*I*_hold_) as previously applied [35,45]. *V*_hold_ values more negative than *V*_thres_ were selected. The presence of an H^+^ selective leak in the deactivated-state was determined by an upward or downward shift in *I*_hold_ driven by the respective driving force (*V*_hold_ − *V*_rev_). Test pulse durations were 1 s, given at 30 s intervals at *V*_hold_, to ensure better control of internal pH_i_ conditions due to either depletion or accumulation of protons. Cells were perfused with more than 5 mL of the testing buffer for periods of at least 30 s.

### 2.9. Statistical Analysis

A statistical significance test was performed to compare the biophysical properties of the mutants with those of the WT hH_V_1 channel. This was achieved using a one-way ANOVA with Tukey’s post hoc test for mean comparison, applying an alpha value of 0.05.

## 3. Results

### 3.1. Somatic HVCN1 Mutations in Cancer: Statistics and Distribution

A total number of 197 somatic *HVCN1* mutations in various cancer types affecting 125 out of 273 residues (45.8%) of hHv1 were incorporated into our dataset after analyzing three different databases (COSMIC, TCGA, and ICGC). As some mutations were found more than once in different cancer patients, this number is reduced to 163 independent mutations (see Appendix A). The total number of somatic mutations included 134 missense mutations, leading to an amino acid exchange on the respective position, 51 sense mutations, and 12 mutations introducing a stop codon leading to a C-terminal truncation. A distribution of the mutations within different tumor types according to the COSMIC database is given in Appendix A. Especially high frequencies of hH_V_1 mutations were found in the liver (3.20%), placenta (3.23%), endometrium (3.94%), and skin (4.71%).

Somatic mutations are not evenly distributed among genes. Depending on the structural features and accessibility of the genomic DNA harboring the respective gene, some regions accumulate more mutations than others [1]. For *HVCN1*, two “hot spots” were identified in the coding region of the central part of the N-terminal region and in the region coding for S4 and the adjacent C-terminal region. In both regions, the number of residues affected is much higher than in other regions: 72.7% of the residues in the central N-terminal region are affected, 53.3% in S4, and 47.6% in the adjacent C-terminal region, compared to <40% in other regions. Similarly, the total number of mutations found per residue is much higher in the “hot spot” regions, with values around 1.0 mutation/residue compared to values around 0.5 mutation/residue in other parts of the gene (summarized in Appendix A).

In Figure 1, the number of missense mutations found within a seven-residue window around a particular amino acid position within hH_V_1 is shown, clearly indicating the two “hot spots” for somatic mutation in *HVCN1*. In addition, in Figure 1, the positions of silent and stop codon mutations are indicated by green and blue dots.

In a recent meta-analysis [46], hH_V_1 residues with important and well-known functions were listed (shown as red dot in Figure 1). Not surprisingly, many of these functionally relevant residues overlap with somatic cancer mutations, especially in the transmembrane regions (indicated by a blue dot in Figure 1). Focusing on the “hot spot” in the S4 transmembrane segment containing the hH_V_1 voltage-sensor, we analyzed some of these mutations electrophysiologically and by MD to gain insight into their potential influence on cancer cell phenotype. In total, nine relevant S4 mutations were identified within the COSMIC database: one R205W (skin/malignant melanoma), two R208W (both in the large intestine/adenocarcinomas), four R208Q (endometrium, large intestine, lung, B cell lymphoma), one G215R (lung/squamous cell-carcinoma), and one G215E (skin/squamous-cell carcinoma) (see Appendix A for more information).

### 3.2. Proton Conduction

All tested mutants exhibited clear characteristics of hH_V_1-driven proton currents (Figure 2A). H^+^ currents increased with cell depolarization, followed by the appearance of typical tail currents at hyperpolarized potentials. The currents were time-dependent. Both the activation and deactivation rates of the mutant channels were different than those of WT. Activation of currents showed an exponential rise after a short delay. This delay has been associated with cooperativity between the two subunits in hH_V_1 homodimers during gating [47] and is more evident in slower-activating channels such as WT, R208W, G215E, and G215R. An initial delay in the rise of H^+^ currents is not resolved in the rapidly activating R205W and R208Q mutants (Figure 2A). In the case of the R205W mutant, a fast (almost instantaneous) and a slow component were seen during activation. At highly hyperpolarized potentials (<−60 mV), the *g*_H_-*V* curves of R205W remain relatively shallow, making the characterization of low-conductive states more difficult.

The tail current is accelerated in all mutants except R208W, which displays exceptionally slow tail currents. Additionally, all the mutants exhibit two-component tails. Furthermore, as in WT hH_V_1 channels [48], proton conduction in all mutants is regulated by ΔpH, as *g*_H_-*V* curves shift positively or negatively along the voltage axis when pH_o_ is decreased or increased, respectively.

The transfection of both tsA201 and HEK cells leads to a WT channel maximal conductance density of 1.6 ± 0.2 nS·pF^−1^ (*n* = 4) (Figure 2B). Compared to the WT hH_V_1, all the somatic mutants showed a lower conductance by at least one order of magnitude, ranging from 0.05 ± 0.01 nS·pF^−1^ for the G215R mutant to 0.6 ± 0.1 nS·pF^−1^ for G215E.

To evaluate whether mutations affect the voltage-dependent gating of hH_V_1, we estimated the number of gating charges (*e*_0_) using the limiting slope strategy [49,50] as previously applied in H_V_ channels [42,44,45,48,51,52,53,54]. In this way, we avoid potential errors that could arise when *e*_0_ is obtained by Boltzmann fittings of normalized *g*_H_-*V* curves [51]. *e*_0_ represents the number of elementary charges that move across the electric field before proton conduction is initiated. In the case of dimeric proton channels, these values have been reported to range from 4.3 to 6.1 e_0_ [55], which may reflect the presence of three arginines in the S4 helix per H_V_1 monomer. The calculated elementary charges are reduced for the mutants in comparison to the WT channel (6.2 ± 0.2 *e*_0_) (Figure 2C). Mutants with substitutions at either the first arginine (R1) or the second arginine (R2) in S4 showed a significant reduction, with values around 2–3 *e*_0_: 2.7 ± 0.5 *e*_0_ for R205W, 2.8 ± 0.2 *e*_0_ for R208W, and 2.3 ± 0.1 *e*_0_ for R208Q. In contrast, mutations of Gly215 to charged residues led to a more moderate reduction in gating charge values, with *e*_0_ values of 5.2 ± 0.3 for G215E and 3.9 ± 0.3 for G215R.

The activation threshold (*V*_thres_) was defined as the most negative voltage at which a characteristic tail current appeared after a depolarizing pulse. As depicted in Figure 2D, WT, R205W, and G215R channels activate similarly at potentials close to 0 mV under symmetrical pH conditions. The G215E mutant activates at a slightly more positive potential (+5 ± 2 mV) than the WT channel (−3 ± 2 mV). However, a noticeable shift to more negative potentials was observed in the mutants with a R2 substitution. Both the R208W and R208Q channels activate at more hyperpolarized potentials: −22 ± 4 mV and −18 ± 2 mV, respectively.

When analyzing *g*_H_-*V* relationships in terms of half-activation voltage (*V*_0.5_) (Figure 2E), the two R2 mutants and G215R resemble WT (+21 ± 3 mV), with values ranging from +21 to +27 mV. Similar to *V*_thres_, G215E displays a positively shifted value of +48 ± 4 mV. R205W exhibits the largest change in V_0.5_, with a shift of nearly +77 mV to more positive potentials. Nevertheless, the weak voltage dependence of *g*_H_-*V* relationships of R205W was poorly fitted by a Boltzmann function, and, therefore, *V*_0.5_ values for this mutant must be taken with caution.

Tested in a pH range from 5.5 to 7.5, all the mutants are proton-selective, with the measured *V*_rev_ matching the predicted Nernst potential for protons *E*_H_ (Figure 2F). Under the experimental conditions, i.e., pH_o_//pH_i_ = 7.5//6.5 and [TMA^+^] = 100 mM as the main cation, the proton permeability for the mutants is ≥3 × 10^6^ P_H_^+^/P_TMA_^+^, which is as high as previously reported in different proton channels [52].

### 3.3. Kinetics

Comparing the kinetic parameters of different H_V_1 channels is challenging because experimental conditions must ensure standardization, including consistent electrochemical forces, similar cell access, temperature, open probability, and other factors. To standardize and compare the kinetic behavior of the mutants, we selected experimental conditions that minimize the accumulation or depletion of protons, which could otherwise alter the protonic driving force (*V* − *V*_rev_) and affect the results. Additionally, we aimed to compare values obtained at the same open probability (see Section 2.7). Specifically, we selected potentials representing half-activation (*V*_0.5_) and −40 mV at ΔpH = 0 to analyze activation and deactivation kinetics, respectively.

In contrast to WT hH_V_1, all somatic mutants analyzed exhibit tail currents with two distinct components. As in WT, currents activate with a single-exponential time course after a short delay in all mutants except R205W. The R205W mutant displays both fast and slow components in its activation kinetics. Moreover, R205W appears to have a weaker voltage-dependence of gating kinetics, especially evident for deactivation kinetics.

Figure 3A shows current traces of WT, a fast-activating (R205W) and a slow-activating (R208W) mutant under the same electrochemical forces. The rising sigmoidal-shaped H^+^ currents characteristic of proton channels is more pronounced in WT and slow-activating channels, such as R208W, whereas activation appears more exponential in fast-activating channels Figure 2A and Figure 3A). Average *τ*_act_ values show greater variability compared to *τ*_deact_ (Figure 3B,C). R208Q (*τ*_act_ = 1.04 ± 0.07 s) and particularly R205W (*τ*_act_ = 0.09 ± 0.02 s) activate faster than the WT channel (*τ*_act_ = 2.0 ± 0.3 s). The remaining channels, R208W, G215E, and G215R, all activate slower than WT, with rates from 1.6 (G215E) to more than 6 times slower (R208W).

On average, all mutants except R208W deactivate faster than the WT channel (*τ*_deact_ = 0.6 ± 0.1 s). The G215E mutant is the fastest, exhibiting a *τ*_deact_ up to six times faster than WT (Figure 3C). Similar to the activation kinetics, R208W differs markedly from the other mutants, as it deactivates much more slowly than WT hH_V_1. The slowing observed in R208W is the most pronounced among all the channels. Both activation and deactivation are affected, with average *τ*_act_ and *τ*_deact_ exceeding 12 and 5 s, respectively. With the exception of the extremely slowly gating R208W mutant, there was no clear relationship between activation and deactivation kinetics.

Substitutions at the neutral residue G215 result in channels that activate more slowly but deactivate at least twice as fast as WT, indicating increased stabilization of the closed configuration. For both activation and deactivation, mutations at the cationic positions R1 and R2 to Trp affect hH_V_1 kinetics in an opposite manner. In comparison to WT, the R2→W substitution slows the kinetics, while R1→W makes it faster. In contrast, the other substitution in R2, R2→Q, accelerates the kinetics. Interestingly, although R2→W and R2→Q activate at notably different rates, they deactivate with comparable time constants, which are also similar to those of the WT channel.

To evaluate the effect of each individual mutation on hH_V_1 kinetics of gating, we calculated the free energy change on the kinetics produced by each amino acid substitution and compared it to the WT channel (see Section 2), following a previously applied strategy [56,57]. This free energy change (ΔΔ*G*) is interpreted as perturbation energy where ΔΔ*G* values further from 0 kcal·mol^−1^ (WT) indicate perturbations in channel kinetics. In our analysis, a negative perturbation (ΔΔ*G* < 0 kcal·mol^−1^) suggests a reduction in the energetic barrier of the *closed* ↔ *open* transition, while a ΔΔ*G* > 0 kcal·mol^−1^ indicates an increase in this barrier.

Regarding activation kinetics, only R205W and R208Q reduce the energetic barrier for the closed-to-open transition, with a particularly notable reduction in R205W (ΔΔ*G* = −1.8 ± 0.2 kcal·mol^−1^) (Figure 3D), which thus facilitates the transition. For the other mutants, the energetic barrier increases compared to WT, ranging from minimal in G215E to mild in G215R. The most pronounced increment on the perturbation energy was observed in the R208W channel (ΔΔ*G* = 1.1 ± 0.1 kcal·mol^−1^).

In terms of deactivation kinetics, a distinct pattern emerges: all substitutions decrease the energetic barrier for the open-to-closed transition, except for R208W, which—similar to its activation kinetics—shows an increased barrier of 1.3 ± 0.1 kcal·mol^−1^ compared to WT. Notably, the G215 mutants exhibit larger negative perturbations (<−0.6 kcal·mol^−1^), while the effects are minimal for R205W and R208Q (~−0.2 kcal·mol^−1^).

We further analyzed the voltage dependence of the gating kinetics by fitting the *τ*-*V* relationships to by linear regression to obtain their slopes (insets from Figure 3A). The voltage dependence of the kinetics (*U_τ_*) was then determined as the reciprocal of these slopes, expressed as mV per *e*-fold change (Figure 3E). Due to the apparent loss of voltage dependence in R205W, only the voltage range showing linearity was included in the analysis. The relationship between the voltage dependence of deactivation and activation kinetics is expressed as the quotient (θ) of *U_τ_*_,deact_ and *U_τ_*_,act_ (Figure 3D).

Our data indicate that WT hH_V_1 has voltage-dependent kinetics of approximately 40 mV per *e*-fold change, which is nearly identical for both activation and deactivation (Figure 3E). A clear difference between *U_τ_*_,*deact*_ and *U_τ_*_,*act*_ is observed in R205W. Although R205W currents activate with a voltage dependence comparable to WT (45 ± 6 mV/*e*-fold), *Uτ*_,deact_ is drastically reduced, showing average values greater than 130 mV/*e*-fold, which is more than three times less voltage-dependent than the activation (Figure 3F).

R208Q and G215R exhibit a mild decrease in *U_τ_*_,*act*_, while R208W and R205W show a value similar to WT. In contrast, G215E has an increase in *U_τ_*_,*act*_, which is twice as high as that for WT (Figure 3E). Among all mutants, only R208Q and G215R display a greater *U_τ_*_,*deact*_ than *U_τ_*_,*act*_ (Figure 3F).

### 3.4. R205W Leaks Protons in the Deactivated State

Natural mutations in the VSM of newly discovered H_V_ paralogs have been shown to cause a proton leak in the closed state [45], similar to the effects observed in in vitro studies of hH_V_1 hydrophobic gasket mutants, where amino acid substitutions disrupt the gasket [35]. Consequently, we aimed to determine whether the somatic mutations in hH_V_1 could induce proton leakage in the deactivated state. To this end, we employed a strategy previously used to study similar phenomena (see Section 2). For each mutant, we exchanged external buffer solutions with varying pH_o_ values during test-pulse protocols to record the direction of currents elicited at the holding potential (*I*_hold_) (Figure 4). In all our experiments, the internal pH_i_ was maintained at 6.5 for all mutants, and we applied pH_o_ changes starting from symmetrical pH_o_//pH_i_ = 6.5 conditions. The holding potential was set in each experiment to ensure channel closure under different pH gradients while monitoring the initial *I*_hold_ (blue dashed line in Figure 4). If a channel leaked protons in the deactivated state, *I*_hold_ would follow the direction of the electrochemical driving force once pH_o_ was changed, resulting in an upward or downward shift (blue arrows in Figure 4). Among all the mutants tested, only R205W exhibited H^+^ leakage in the closed state.

### 3.5. Results of MD Simulations

Any single amino acid mutation at the sequence level may lead to subtle structural variation but pronounced functional differences in the mutant as compared to the WT hH_V_1. Therefore, we explored the dynamics of the mutant channels and compared them to the WT to address the effects of mutations on structure and function.

#### 3.5.1. MD Trajectories Analysis

With the exception of the mutated side chains, the starting structures for the MD simulations were the same for all WT and mutant channels, either for the deactivated and the activated configuration. Thus, the starting structures of the deactivated and activated channels are optimally suited as a reference to compare the time evolution of the structure of the different channels. To assess the overall stability of the WT and mutant channels during MD simulations, we calculated the protein backbone Root Mean Square Deviation (RMSD) and Root Mean Square Fluctuations (RMSFs). Details about the calculations are given in the Appendix A. The results presented below are shown graphically in Appendix A.

-RMSD

The RMSDs generally deviated from the starting structures in a range from 2 to 4 Å for all the channels in both configurations, as illustrated in Appendix A. Only G251R in the deactivated configuration and R208Q in the activated configuration made a few escapades beyond 4—but still below 5 Å—and these remained relatively ephemeral. Considering the part of the simulation after 3 µs, the RMSDs did not deviate by more than 1–1.5 Å anymore, remaining stable at around 2.5 to 3.5 Å and indicating that the channel backbones were stable after this time, as emphasized by the narrow probability distributions of the backbone RMSDs during the last 2 µs of the simulation in Appendix A.

-RMSF

The calculated RMSFs confirm that for all the channels, the N- and C-termini and the loops that connect the transmembrane helices are the most flexible regions, as expected. In both the deactivated and activated configurations, all the mutants fluctuate close to the WT channel. Only one larger exception is observed for deactivated G215R. Also, larger (but smaller than for G215R) deviations are observed for the deactivated R208 mutants too, but they are located essentially at the C-terminus for R208Q and at the N-terminus for R208W and thus do not affect the transmembrane domain itself. In the activated configuration, only R208Q deviates slightly from WT, especially in the S1 helix.

-Clustering

To identify the different conformational states sampled by the channels during the MD simulations, the snapshots collected along the trajectory were grouped into clusters of similar protein backbones. Details about the calculation are given in the Appendix A and the results of the clustering are summarized in Appendix A and in Appendix A.

For all the channels, the most populated cluster is the cluster populated in the last part of the trajectory, c0 in Appendix A and in Appendix A. Only for deactivated R205W, the cluster populated in the last part of the simulation is the second most populated cluster, c1. Since the c0 clusters (c1 for deactivated R205W) are furthermore populated in a significant portion, always more than 30% (with only one exception for deactivated G215R), the structures in these clusters were used for further analyses.

#### 3.5.2. Structural Features of WT and Mutant Channels

In a previous work, we investigated the effects of pH on WT hH_V_1 and identified structural features that distinguish the pH-deactivated and pH-activated states [36]. Especially, we found a network of stabilizing salt bridge interactions in the pH-activated channel that extends from the intracellular entrance to the HG and from the HG to the extracellular entrance of the channel. This extensive network is missing in the pH-deactivated channel, which exhibits rather localized salt bridges. We also identified an outward displacement of the transmembrane helix S4, by approximately one helix turn, that passes the second voltage-sensing arginine R2 through the HG during activation. Furthermore, we found that the HG and the rest of the channel crevice are more open in the activated configuration.

-Salt bridge interactions

Details about the calculations of the salt bridges are given in the Appendix A. The probability densities of the salt bridges among the structures in the c0 clusters (c1 for deactivated R205W) are shown in Appendix A. For more clarity, the significant salt bridges are listed in Table 1 and shown schematically in Figure 5. A graphical representation of the clusters’ centroid structures with the residues presented below is shown in Appendix A. The salt bridges identified for the WT channel are very similar to those found for the pH-deactivated and pH-activated states in our previous work [36]. The deactivated configuration has several clusters of tight but “localized” salt bridges. Especially, R1 forms a tight salt bridge with D112 in the external vestibule, above the HG, and R2 and R3 form strong salt bridges with D174 and with E171, respectively, in the internal vestibule below the HG. Neither D112 nor D174 are engaged in further salt bridges, but E171 interacts with K157 too. K157 itself interacts with E153 so that R3 is engaged in a larger cluster of salt bridges. A second local “cluster” of salt bridges involves K131 with D123 and E119 at the extracellular entrance of the pore. These interactions are lost in the activated configuration, suggesting that the extracellular entrance might be wider, in agreement with our previous results [36]. In the activated configuration, R1 interacts tightly with E119 and with D123 at the extracellular entrance. R2 interacts both with D185 and with D112 in the external vestibule. D185 interacts further with H140, extending the network of interactions from the selectivity filter D112, directly above the HG, up to H140 that is located close to the extracellular entrance. In the internal vestibule, R3 interacts with D174, located directly below the HG. However, it does not participate anymore in the cluster formed by E171, K157, and E153, in contrast to deactivated WT.

The salt bridges involving the putative voltage-sensing arginines R1, R2, and R3 in helix S4 are generally comparable in all the channels, both in the deactivated and activated configurations. However, subtle differences are specific to each channel. Due to the mutation at R2, a salt bridge with D112 is missing in the activated configuration of the R208Q and R208W mutants. However, the oxygen atom of the side chain amide group enables the glutamine in R208Q to form an interaction with D185, partially recovering the R2-D185 salt bridge as in WT. Similarly, the salt bridge with D174 in deactivated WT is missing in deactivated R208W, but a weak interaction between R2 and the R208Q is formed via the side chain amide group. The interaction between H140 and D185 in the activated configuration is lost completely in the R208W mutant only. Remarkably, with only five tight salt bridges, R208W has as few tight salt bridges as R205W, missing two to four strong stabilizing interactions compared to the other channels. Due to the R→W mutation at R1 in R205W, the interactions of R1 with D112 and D185 in the deactivated configuration are lost. Actually, there is no salt bridge in the external vestibule of deactivated R205W. However, an interaction with D112 is somehow “recovered” via R2’s side chain. This interaction takes place across the HG and is not due to the displacement of R2 above the HG (see the subsection on “R2 and S4 shifts”). Thus, unlike all the other channels, an interaction between R2 and the selectivity filter D112 is readily formed in the “deactivated” configuration of R205W. This not only helps to recover the missing interaction of D112 with R1 but, much more remarkably, it creates a linkage between the external and internal vestibules across the HG (see the subsection on “Complementarity of electrostatic potential and water wire”). Strikingly, the same mutation at position R2 (one helix turn after R1) in R208W induces a similar effect in the activated configuration. After S4 has moved outwardly by approximately one helix turn, displacing Trp208 approximately at the position of Trp205 in deactivated R205W, an interaction across the HG is formed between R3 and D112, although it is weaker in R208W. Otherwise, the patterns of interactions in the internal vestibule of R205W are comparable to those in all the other channels in the two configurations. However, in activated R205W and R208Q, R3 is also located above the HG, in the external vestibule (see the subsection on “R2 and S4 shifts”). Consequently, and unlike the other channels, R2 interacts with E119, instead of R1 in the other channels, and R3 interacts with D112, both in R205W and in R208Q.

Briefly, the salt bridges found in WT are nearly conserved in all the mutants. Regarding tight interactions, R205W has fewer of those than all the other mutants in both configurations. Strikingly, the same R→W mutation at R2 in R208W also leads to fewer tight interactions in both configurations. In contrast, the mutation of R2 to a polar glutamine (in contrast to the nonpolar Trp) in R208Q does not perturb the number of tight salt bridges compared to WT, at least in the deactivated configuration. Altogether, the biggest effects of a mutation on the structure are observed for the R→W mutation in R205W.

-R2 and S4 shifts

The shifts of the second voltage-sensing arginine (R2) in helix S4 along the transmembrane axis and of the S4 helix itself between the deactivated and activated configurations are shown, respectively, in Appendix A. Details about the calculations are given in the Appendix A.

As depicted in Appendix A, and considering the thickness of the HG in the z-direction, the second voltage-sensing arginine Cα atom and side chain are located below or in the HG in all deactivated channels, whereas they are located above the HG in all activated channels. In the activated configurations, R2 is located higher above the HG in all the mutants compared to WT, whereas in the deactivated configurations, the mutants are distributed around WT. The mean value, among all WT and mutants, for the shift of R2 between the deactivated and activated configurations calculated from all the WT and mutant channels is 7.0 ± 1.5 Å (7.0 Å in WT), considering the backbone Cα atom, and 8.4 ± 1.9 Å (7.5 Å in WT), considering the side chain. Comparing the Cα and side chain positions for one given channel, it appears that the side chain is generally slightly lower than the backbone in deactivated channels but slightly higher in activated channels, indicating that R2’s side chain is generally orientated toward the intracellular cytosol in deactivated and toward the extracellular solution in activated channels, respectively. In deactivated R205W, however, R2’s side chain is higher than Cα and is consequently orientated toward the external vestibule.

The calculated shifts for the S4 helix vary between ~3.8 Å for G215E and ~7.5 Å for R205W. S4 shifts are generally of smaller amplitude than those of R2, by 1–3 Å (based on R2 Cα) depending on the channel. In most channels, the shift corresponds almost to one helix turn of an alpha helix (5.4 Å), ±0.7 Å: WT 5.0 Å, R208Q 4.7 Å, R208W 5.9 Å and G215R 5.4 Å. Only R205W exhibits a larger shift of 7.5 Å.

-Electrostatic potential

The structural features analyzed previously have already revealed obvious features specific to the R205W mutant. It has the lowest number of stabilizing interactions in both deactivated and activated configuration. More strikingly, the second voltage-sensing arginine, R2, already interacts with the selectivity filter aspartate, D112, across the hydrophobic gasket in the deactivated configuration. This feature is unique to R205W. R205W has one of the largest R2 and S4 helix shifts between the deactivated and activated configurations, 1.5 and 2.3 Å more than WT for R2 and S4, respectively. Finally, in the patch-clamp experiments, R205W exhibits a proton leak, meaning that it conducts protons across the membrane under conditions where a non-conducting, deactivated channel is normally expected.

Therefore, in the next step, we looked at the electrostatic potential in the “deactivated” configuration of the R205W mutant and, for comparison, in the deactivated and activated configurations of the wild-type channel. The electrostatic potential maps of the channels are shown in Figure 6, together with the 3D potential isosurfaces at the isovalue of −160 mV. Details about the calculation and additional figures are given in the Appendix A. In deactivated WT, the electrostatic potential vanishes at the HG, as revealed by the white region at the HG and emphasized by the discontinuity of the isosurface at −160 mV, resulting in a non-attractive region for protons. In contrast, in “deactivated” R205W, the electrostatic potential remains continuous throughout the HG, as it is in activated WT, resulting in an attractive force that expands through the whole channel, from the intracellular cytosol to the extracellular matrix.

-Water number along *Z*-axis

Another feature that we analyzed to compare the WT and 205W channels is the number of water molecules along the pore axis, orthogonal to the membrane. Details about the calculations and a figure representing the results (Appendix A) are given in the Appendix A. In deactivated WT and R205W, the HG is dewetted (no water molecule at z = 0 in Appendix A). Additional dewetted regions in most of the structures analyzed are the region between R1 and D112 in WT (z ≈ 5 Å) and the region between D185 and E119 in R205W (z ≈ 12–15 Å). In contrast, the water wire is continuous through the whole pore axis of the activated WT channel, although it is dewetted at the HG in approximately half of the structures, as shown by the water number of ≈0.5 at this location. Compared to deactivated WT, the two regions directly below (z ≈ −10–−3 Å) and above (z ≈ 3–10 Å) the HG are filled with more water molecules in R205W. But especially, the dewetted region in-between is slightly narrower in R205W than in WT, with 3.5 Å against 4.5 Å, respectively, as measured for a water number of 1. The distance between two water molecules at the HG in R205W is thus close to the average distance between water molecules in bulk solution, 3.1 Å, suggesting that proton hopping across the HG in the “deactivated” configuration of R205W might be possible. Supporting this idea, the water represented as a surface in the centroid structure of R205W’s cluster c1 is almost continuous at the HG, as shown in the right panel of Figure 7A.

-Complementarity of electrostatic potential, and water wire

Figure 7 superimposes the electrostatic potential isosurface at isovalue = −160 mV together with the water surface in the centroid structures of the analyzed cluster c0 for WT and c1 for R205W. As shown in this figure, both the electrostatic potential and the water wire are discontinuous in deactivated WT. In cluster c1 of R205W, either the electrostatic potential or the water wire, or both at the same time, are continuous along the whole “pore” axis in almost all the structures of the cluster.

## 4. Discussion

H_V_1 has been related to several cancer types [16,26,27,28,29,30,58] while pH homeostasis linked to cancer development [23,24,25]. We have analyzed the impact of these mutations on the biophysical properties of the ion channel, which will be further discussed here. *HVCN1* is located on chromosome 12, which has two copies in a human cell. Thus, the mutation of just one locus might be compensated by the other. Interestingly, especially in cancer biology, a loss of heterozygosity (LOH) is a very common event and loss of the wild-type allele during tumor progression is not rare [59]. Therefore, the allele with the mutation frequently persists, dominating the structure and function of the channel in the cell. Deleterious passenger mutations inhibit tumor cell proliferation [6]. All the somatic mutations analyzed here compromise the function of hH_V_1, as shown by electrophysiological measurements and by the computation of the presumed structures. pH homeostasis is key in cancer development, and the results suggest that the mutations analyzed may compromise tumor proliferation.

Arginine mutations are more commonly observed than other mutations in cancer, especially arginine-to-histidine substitutions [60,61]. Three of the mutations investigated here include substitutions of cationic Arg residues. In hH_V_1 S4, Arg→His substitutions were not found. Instead, the voltage-sensing arginines R1 and R2 are both mutated to Trp. Arg-to-Trp mutations are frequent in driver mutations present in endometrial or breast cancer [56]. The Arg→Trp mutants in this study were found in intestine and skin cancer, while the Arg→Gln mutation was found in endometrial, intestinal, lung, and hematopoietic tumors (Appendix A). Mutations of Gly, as found here for hH_V_1 at G215, are rather uncommon in cancer. Thus, our findings are consistent with the pre-existing literature. However, the low number of documented hH_V_1 somatic mutation cases prevents more specific conclusions.

### 4.1. All Mutants Reduce the Effective Charge Displacement

Similar to other voltage-activated proteins, the VSM of the hH_V_1 channel contains a series of basic residues arranged in a unique form RxWRxxR (Figure 8). Among the mutations studied, three affect the first two charged arginine residues (R1 and R2) of the VSM. The R→W and R→Q mutations both neutralize positive charges of the voltage sensor. At symmetrical pH conditions, the total effective charge measured is reduced by more than 3 *e*_0_ in each mutant (Figure 2C), where one would expect 2 *e*_0_ per dimeric unit if each VSM arginine contributed 1 *e*_0_ to the total charge displaced. A reduction of more than 2 *e*_0_ when a single VSM arginine was mutated to non-charged amino acids has been described previously. For example, neutralization of R1 reduces the effective charge by ~37% in human and ~66% in *Ciona* H_V_1 [51,62], while neutralization of the second arginine, R2, to asparagine reduces *e*_0_ by ~50% [51]. On the other hand, mutations that introduce a charge at position 215, just below the VSM, lead to small (1 *e*_0_) or mild (2.3 *e*_0_) gating charge reductions for G215E and for G215R, respectively.

The reduction in *e*_0_ from its nominal value (6 *e*_0_ for dimeric hH_V_1) might arise from different scenarios. These scenarios might happen either individually or in combination, especially in cases where the charge reduction is more than expected: 1. elimination of the charge per se; 2. restriction of the movement of charge carriers, lessening the amount of total charge crossing the electrical field; 3. modification of the channel structure, which compromises its voltage-dependence; 4. changes in dimerization probabilities or cooperativity between the hH_V_1 protomers; 5. changes in electrodynamics of the channel.

### 4.2. Mutations of Voltage-Sensing Arginines Impact hH_V_1 Kinetics in Distinct Ways

Among the mutants studied, R205W (R1→W) and R208W (R2→W) exhibit the most significant functional differences. R1→W activates the fastest, within a voltage range similar to WT, while R2→W activates the slowest and at more negative voltages (Figure 2D). Strikingly, the voltage dependence of activation kinetics for both mutants is comparable to WT. The effects of the R→W substitution may be attributable to the bulky size and unique chemical properties of tryptophan. For instance, it is well known that the indole group of tryptophan contributes to π- and hydrophobic interactions with chemical groups on other residues [63,64,65,66], which play crucial roles in protein stabilization and anchoring [67]. Furthermore, the conserved Trp207, also located in the VSM and common to all H_V_ channels, is responsible for the slow opening and closing kinetics of hH_V_1, promotes the closed state, affects ΔpH gating [68], and contributes to the energy barrier during the transition from closed to open states [69].

The R1→W and R2→W channels display the fewest electrostatic interactions and the largest S4 shift (Appendix A). During the transition from deactivated to activated states, S4 in R1→W and R2→W shifts, respectively, 1.0 Å and 2.5 Å more than in WT, with R1→W exhibiting the largest shift (Figure 8). Since the two mutants exhibit opposite kinetic effects, these results suggest that channel kinetics are independent of S4’s displacement or/and of the number of electrostatic interactions.

The size of the substituted side chain may influence channel kinetics, as tryptophan has the largest side chain among the 20 natural amino acids. A study of HERG channels reported that tryptophan substitutions in the S4 region increased side chain volume by up to 96 Å^3^, requiring volume correction when estimating perturbation energies [70]. Consistent with the idea that bulky residues have a greater impact on channel dynamics, the kinetic analysis shows that the R1→W and R2→W mutations generate the highest energy pattern perturbations and lead to substantial changes in channel structure (Figure 3 and MD analysis). Although we did not adjust for side chain volume in our kinetic analysis, the Arg-to-Trp substitutions at positions 205 and 208 have opposing effects: R1→W accelerates activation, while R2→W slows it down. In the case of R1 position, previous substitutions to Ala in human H_V_1 and to Gln in mouse H_V_1 also produced an acceleration of the activation kinetics [62,71], coincidently with the dynamics seen in R1→W. In R2, the tested mutations have two contrasting effects, since R2→W increases *τ*_act_ but R2→Q diminishes it (Figure 3). Interestingly, mutation of R2 in human H_V_1 to Ala decreases *τ*_act_ also [62], similarly to Gln substitution.

This study does not allow us to determine a correlation between the size of the substituents at R1 and R2 and the behavior of the activation kinetics. However, the contradictory kinetic effects of Trp substitutions at R1 and R2 might be related to the chemical nature of Trp and its specific interactions with the surrounding environment during gating. The location of R1 in the external vestibule in the deactivated state [36,39,51,72] implies that during gating, this residue moves exclusively within the external vestibule of the channel, which becomes more open from the HG towards the extracellular environment. In the deactivated WT channel, R1 interacts with D112 located just above the HG in the outer vestibule, while R2 interacts with D174 in the internal vestibule (below the HG). During gating, the S4 alpha helix shifts outward. This displacement brings R1 in close, stabilizing contact with E119 and D123, close to the external entrance of the channel, while R2 crosses the HG in a *one-click mechanism* and forms a stabilizing interaction with D112 in the active configuration (Table 1 and Figure 8). The R1-D112 interaction in the deactivated WT channel is lost in the deactivated R1→W channel and is replaced by a new interaction between R2 and D112 that takes place across the HG. In this configuration, Trp205 is located above the HG, in the external vestibule. Upon gating, Trp205 moves to approximately the same height as the external E119 and D123 cluster, but the mutation eliminates potential interactions with these residues. Instead, Trp205 moves towards the lipidic membrane (Figure 7B). The two remaining S4 arginines, R2 and R3, cross the HG in a *two-click mechanism*. In this active state, R2 interacts essentially and strongly with E119, while R3 becomes the essential interaction partner of D112, taking the position typically occupied by R2 (Table 1 and Figure 8). According to our MD simulations, the trajectory of Trp205 occurs exclusively in the external water vestibule. In both configurations, the Trp205 side chain is located almost at the interface between the channels’ core and the lipid membrane (Appendix A and Figure 7B). Although we did not simulate the deactivated-to-activated transition, this observation almost rules out strong interactions between Trp205 and other amino acid residues within the pore of the channel that may stabilize the S4. It has been demonstrated that Trp substitutions in S4 can create coupling interactions in the interface between the two H_V_1 protomers [57], where effective Trp-Trp coupling reach ΔΔ*G* values > 1.5 kT [56,57,70]. Our kinetic analysis also excludes this possibility. The perturbation energy estimated for the R1→W channel shows a reduction in ΔΔG = −1.8 kcal/mol (~−3 kT) compared to WT (Figure 3), which is a decrease of more than two times the minimum required for the coupling energy (0.89 kcal/mol). Our data suggest that Trp205 movement is not limited by Trp-Trp interactions during gating, and that the R1→W mutation favors the channel’s active configuration.

In contrast, the Trp substitution in R2 faces a different scenario. In the deactivated state, R1 maintains a strong interaction with D112 above the HG, while the mutated Trp is positioned close to and below the HG, in the internal vestibule of the channel (Figure 8 and Appendix A). In the activated state, Trp relocates to the external vestibule. During the deactivated-to-activated transition, a new R3-D112 interaction across the HG is formed, similar to the R2-D112 interaction seen in the deactivated R1→W channel. The *one-click* transition of position 208 from the internal to external vestibule is supported by several studies [36,38,39,51,73]. Moreover, in both configurations, Trp208 always points towards the channel’s core. Thus, the upward movement of the S4 alpha helix during gating requires Trp208 to cross the HG, moving from the internal to the external water vestibule (Figure 8 and Appendix A), where interactions between Trp208 and HG components likely occur as the channel gates. The MD simulations show close proximity between Trp208 and the HG. Trp’s indole group, located directly below (deactivated channel) or above (activated channel) the HG, points towards the channel’s inside and is almost parallel to the plane formed by the HG’s side chains (Appendix A). The close proximity (4–5 Å) and orientation (40–80°) permits potential π-interaction between the imidazole ring of Trp and the aromatic ring of Phe150. π-π interactions stabilize structures involving aromatic groups [63,74,75], with benzene dimer interactions producing energies ranging from −1.48 to −2.48 kcal/mol [76]. Additionally, weak van der Waals interactions between Trp208 and the HG residues may further contribute to Trp’s stabilization in the region of the HG. These interactions of ~1 kcal/mol are effective at short range and fall beyond a distance of 4 Å [77]. This effect could slow the rate of the activation and deactivation of the R2→W channel, stabilizing the channel in the closed state, as emphasized by its perturbation energy (Figure 3).

Altogether, the results suggest that R1→W and R2→W substitutions lead to distinct gating kinetics due to differences in structural and energetic interactions. R1→W favors activation, while R2→W stabilizes the channel’s closed state by interacting with HG residues.

### 4.3. R205-D112 Interaction Is Dominant in the Deactivated State and Essential to hH_V_1 Voltage Sensitivity

The wild-type hH_V_1 channel exhibits a salt bridge interaction between Asp112 and R205 in the deactivated configuration. This interaction occurs in the external vestibule, above the HG, while a salt bridge is formed in the internal vestibule between R2 and D174. This arrangement positions the HG between the two arginines R1 and R2 (Figure 8).

All the mutants here retain the D112-R1 interaction in the deactivated state, except the R1→W variant. In the deactivated form of R1→W, the side chain of Trp205 remains positioned above the HG but is oriented upward (Appendix A). This orientation prevents interactions with the HG residues that could confer stability to the S4 alpha helix, as discussed above. The loss of the D112-R1 interaction in the R205W mutant leads to a reorganization of R2 and R3 salt bridges. Consequently, the typical R2-D174 interaction seen in WT is replaced by a new R2-D112 interaction across the HG (Figure 8), while D174 interacts with R3. Neutralizing D174 causes a negative shift in the *g*_H_-*V* curves compared to WT, indicating the importance of this residue in stabilizing the deactivated state [78,79].

In the deactivated R1→W channel, the loss of both the external D112-R1 and internal D174-R2 interactions results in an upward orientation of the R2 side chain, which enables an interaction with D112 across the HG, resulting in an electrostatic bridge across the hydrophobic region (Figure 6).

The R1→W mutant has the fastest opening kinetic among the tested variants and is also the only one to exhibit a significant reduction in the voltage dependence of the deactivation kinetics (Figure 3) and proton leakage in the deactivated state (Figure 4). Interestingly, similar traits—including accelerated activation kinetics, reduced voltage dependence of kinetics, and proton leakage in the closed state—have been observed in the third type of H_V_ channel of *Aplysia californica* (AcH_V_3). In AcH_V_3, R1 is naturally replaced by leucine, and proline follows this residue, resulting in an unconventional LPWR2xxR3 signature sequence in the VSM [45]. Proton leakage in the closed state has also been reported when the polarity of the HG is increased, which further revealed a deeper closed state with a slow and weak voltage-dependent component [35]. Similarly, the H^+^ currents conducted by the R1→W channel also shows two kinetic components for both activation and deactivation, suggesting the presence of deeper closed states. Moreover, R205W H^+^ currents are comparable to the currents evoked by AcH_V_3 [45].

Nonetheless, proton leakage in the deactivated form of R1→W may be explained through our electrostatic potential analysis. By comparing the electrostatic potential maps of WT and R1→W, a significant difference emerges. In the deactivated WT channel, the proton-attractive potential is disrupted at the HG, along with a discontinuity in the water wire (Figure 6 and Figure 7). In contrast, when the D112-R1 energy barrier is eliminated in the R1→W deactivated state, water discontinuity at the channel core remains but the proton-attractive potential extends toward the HG, resulting in a continuous potential between the external and internal vestibules, as in the activated WT channel (Figure 6 and Appendix A). The continuity of the proton-attractive potential, together with the interaction between D112 and R2 across the HG, would permit protons to move from one vestibule to the other once an electrochemical change occurs, and it might be responsible for the leakage observed in this mutant.

### 4.4. Intricate Dependence of Voltage Dependence and Charged Amino Acids on S4

The outward movement of the S4 alpha helix during H_V_1 gating is widely accepted to be driven by the voltage-sensing arginines [51,80]. For S4 to shift outward in response to membrane depolarization, the salt bridges that stabilize the channel in the deactivated state must be broken or weakened. Once these interactions are disrupted, the alpha helix can move to a new position, where it forms new interactions that stabilize the channel in a conductive conformation. The breaking and formation of these salt bridges depend on both the pH and electrostatic conditions [36,80].

We analyzed all the channels at symmetrical pH_o_ = pH_i_ = 6.5 to separate the influence of protonation of acidic residues on salt bridge disruption from purely electrostatic effects. Introducing net positive or negative charges at position 215—located in four amino acids below R3—presents an intriguing opportunity for exploration. The G215R mutation introduces an additional positively charged arginine (R4) at the base of the S4 helix. This new arginine might potentially sense membrane depolarization, similar to other positively charged residues in voltage-activated proteins (Figure 8), thereby increasing the total charge displacement. However, our results indicate that the Gly→Arg mutation neither enhances total charge displacement nor promotes additional outward movement of S4. Instead, the addition of arginine at this position reduces the *e*_0_ charge displacement by 37%, suggesting that Gly215 is outside the electric field. Furthermore, adding R4 at position 215 does not increase the voltage dependence of activation kinetics; in fact, it slightly reduces it (Figure 3E). Our data suggest that, rather than facilitating channel opening, charge transfer through the electric field is reduced, likely due to increased stabilization of the deactivated conformation. Despite maintaining electrostatic interactions similar to those in the WT channel in the deactivated state, we observe an additional interaction between R2 and the countercharge at E153 (at the bottom of the S2 alpha helix). Simultaneously, a new interaction of R2 and the counter charge at E153 is observed and the weak R1-D185 interaction seen in the WT becomes stronger (Figure 5 and Appendix A, Table 1). This might globally increase the stability of the S4 backbone and the charge carriers, making the transition from the closed to the open state more energetically demanding.

The preference of G215R for the deactivated configuration is further supported by its kinetics. Along with G215E, this mutant activates more slowly and deactivates more quickly than WT (Figure 3B,C). Kinetic analysis reveals that G215R has a higher energetic barrier for opening and a lower barrier for closing (Figure 3), indicating a promotion of the deactivated state. Similarly, the G215E mutation also favors the deactivated state, slowing activation and accelerating deactivation. The G215E channel activates at more positive voltages than WT, as shown by *V*_thres_ and *V*_0.5_ values (Figure 2). Like G215R, G215E strengthens the R1-D185 interaction in the deactivated conformation. In the activated G215E channel, new electrostatic interactions occur between Glu215 and residues such as R3 or R223 (near the predicted C-terminal coiled-coil) (Table 1 and Figure 5). The G→E mutation also locks the S4 helix closer to the HG in the deactivated state (Appendix A).

Although G215E shows reduced overall S4 shift compared to WT (Figure 8), the total charge moved during gating is at least 1 *e*_0_ higher than that of G215R but still 1 *e*_0_ lower than WT. Of all the studied variants, G215E shows the least alteration in charge displacement through the electric field, interpreted as *e*_0_. Interestingly, G215E increases the voltage dependence of activation kinetics two-fold compared to WT (Figure 3E) and exhibits the smallest upward shift of S4 (Figure 8).

In our MD simulations, we considered the deactivated and activated states separately but did not simulate the pathway between the two states. Surprisingly, the results of substituting a non-polar and minimal glycine residue with charged side chains (Glu and Arg) contradict the expectation that an extra positive charge in S4 would increase voltage sensitivity. The addition of these charged residues creates new interactions within S4 and with countercharges on other transmembrane alpha helices. Additionally, the G215x mutants alter salt bridge partnerships, including those involving the S4 voltage-sensing arginines. Altogether, introducing a net charge at position 215 might impact hH_V_1 voltage sensitivity in an electrodynamic manner, promoting the deactivated state. Further experiments are needed to explain the effects of G215x mutants during gating, particularly considering the cooperativity between hH_V_1 protomers.

## 5. Conclusions

This study provides an in-depth analysis of a somatic mutation hotspot within the S4 segment of hH_V_1, based on data from tumor patients across various cancer types. All the investigated mutations impair the ability of hH_V_1 to extrude protons, potentially disrupting cellular pH homeostasis. Specifically, the R205W mutation generates a proton leak current, which in the acidic tumor environment could increase the cell’s energy demands. The two R208 mutations cause a negatively shifted activation threshold, leading to proton influx in cells exposed to an inward proton gradient. As a result, these mutations are expected to promote proton influx rather than extrusion. The G215 mutation compromises channel function by increasing the energy required for channel opening, thereby reducing proton extrusion and lowering intracellular pH_i_. Since pH homeostasis is critical for cancer cell proliferation and division, these mutations may undermine this essential process. All somatic mutations examined here produce less current per unit of membrane area, suggesting reduced proton extrusion and impaired channel function. Previous research has shown that hH_V_1 expression is upregulated in tumors and contributes to invasive cancer behavior. Our findings indicate that these somatic mutations negatively impact hH_V_1’s normal proton extrusion function, potentially affecting tumor development by disrupting pH homeostasis. However, further studies specifically designed to assess the effects of these mutations on cancer development are still needed.

## Figures and Tables

**Figure 1 biomolecules-15-00156-f001:**
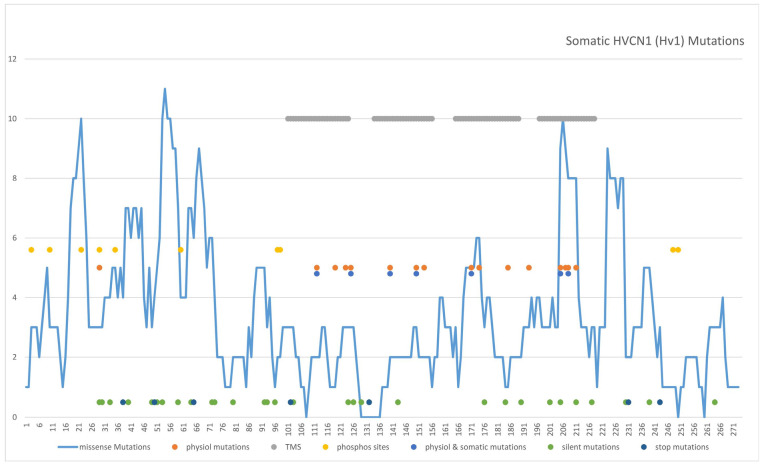
Distribution of somatic cancer mutations in *HVCN1*. For each residue, a “7-amino-acid window” with the number of mutations at the respective site and three residues up- and downstream is shown (blue line), indicating two “hot spots” of mutations: (1) central N-terminal part and (2) S4 and adjacent C-terminal region. Transmembrane regions are indicated as gray bars S1, S2, S3 and S4 (from left to right); putative phosphorylation sites (yellow dots), physiological important residues defined by DeCoursey et al., 2016 [46] (orange dots), and overlapping positions of physiologically important residues also harboring somatic mutations (light blue dots) are indicated. In the lower panel, silent mutations (green dots) and stop mutations (dark blue dots) are given.

**Figure 2 biomolecules-15-00156-f002:**
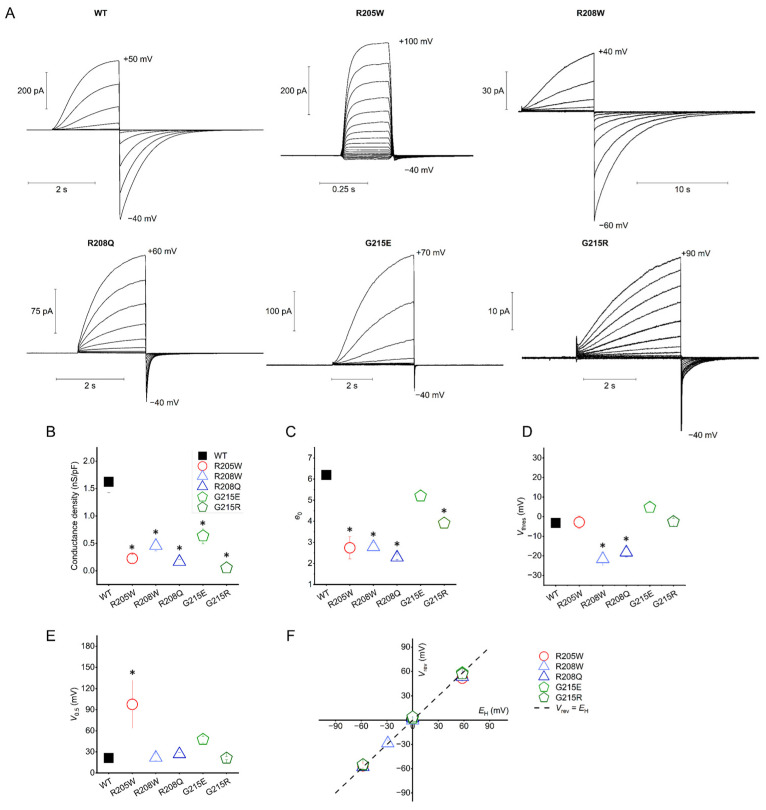
Comparison of hH_V_1 somatic mutants with WT channels. (**A**) Whole-cell recordings of proton currents elicited by the different mutants and the wild-type hH_V_1 (WT). Families of pulses were obtained by depolarization of the cell membrane in 10 mV steps, from the holding potential (*V*_hold_) to the potential shown in each family. For the R205W and G215R families of pulses, depolarization was applied from −80 mV and −60 mV, respectively. *V*_hold_ was −40 mV for all mutants, except for R208W, where it was −60 mV. For all the channels, the families shown were measured at symmetrical pH_o_ = pH_i_ = 6.5. For comparison with same time and current scales, please see Appendix A. (**B**) Maximal conductance density; *n* = 4–7. (**C**) Gating charge obtained by the limiting slope method; *n* = 5–8. (**D**) Threshold of activation (*V*_thres_) of the different mutants in comparison to WT channel measured as the most negative potential where the characteristic tail current appears after depolarization; *n* = 5–8. (**E**) Mean *V*_0.5_ values obtained by fitting normalized *g*_H_-*V* curves to a Boltzmann function; *n* = 4–6. (**F**) Reversal potential (*V*_rev_) vs. Nernst potential for protons (*E*_H_) plot. Mean *V*_rev_ values ± S.D. for the different mutants were measured at different ΔpH = pH_o_ − pH_i_ gradients (ΔpH = 1, 0.5, 0, and −1). For all pH gradients, *n* was between 3 and 7, except at ΔpH = 1, where *n* = 2 for R205W, R208Q, and G215R. In (**B**–**E**), values are shown as mean ± S.E.M and the legend in B is common to all the graphs. Asterisk (*) denotes data where *p* < 0.05 in comparison to WT.

**Figure 3 biomolecules-15-00156-f003:**
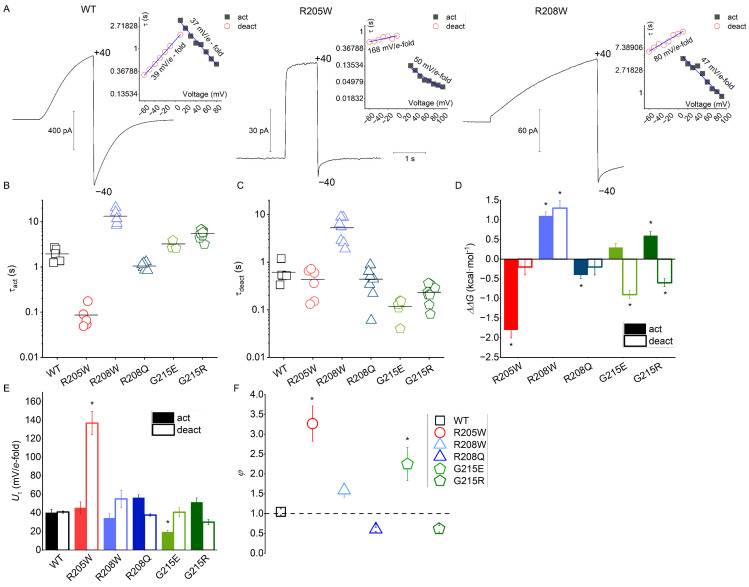
Kinetic analysis. The time constants of activation (τ_act_) and deactivation (τ_deact_) were obtained by fitting to a single exponential current generated by depolarizing pulses and by tail-current protocols, respectively. (**A**) Examples of activation rates at +40 mV for fast (R205W), slow (R208W), and wild-type (WT) channels. For each recording, the holding potential was −40 mV, and the pH was symmetrical at 6.5. Insets show the activation (solid squares) and deactivation (open circles) rate constants as a function of voltage in a range from −60 to +100 mV, where the voltage dependence of kinetics is represented by the linear fitting (blue line), expressed as mV per *e*-fold change. The temporal scale is the same for the three recordings. (**B**,**C**) show the comparison of the time constants of activation at *V*_0.5_ (**B**) and deactivation at −40 mV (**C**) between WT and the different mutants under symmetrical pH_i_ = pH_o_ conditions; *n* = 5–8. Lines in scatter plot represent mean values. (**D**) Perturbation energy of activation (solid bars) and deactivation (open bars) kinetics of the different mutants WT channel (solid vertical line at ΔΔ*G* = 0 kcal·mol^−1^) expressed as ΔΔ*G* (*G*_WT_ − *G*_mut_), at *V*_0.5_ (τ_act_) and −40 mV (τ_deact_) under symmetrical pH conditions (ΔpH = 0); *n* = 5–8. (**E**) Voltage dependence of kinetics (*U*_τ_) of activation (solid bars) and deactivation (open bars) obtained as the slope of *τ*-*V* relationships (i.e., insets in A); *n* = 5–7. (**F**) Ratio *θ* between the slope of the voltage dependence of deactivation kinetics (*U_τ,deact_*) and the slope of the voltage dependence of activation kinetics (*U_τ,act_*); *n* = 5–7. Values are shown as mean ± S.E.M. Asterisk (*) denotes data where *p* < 0.05 in comparison to WT.

**Figure 4 biomolecules-15-00156-f004:**
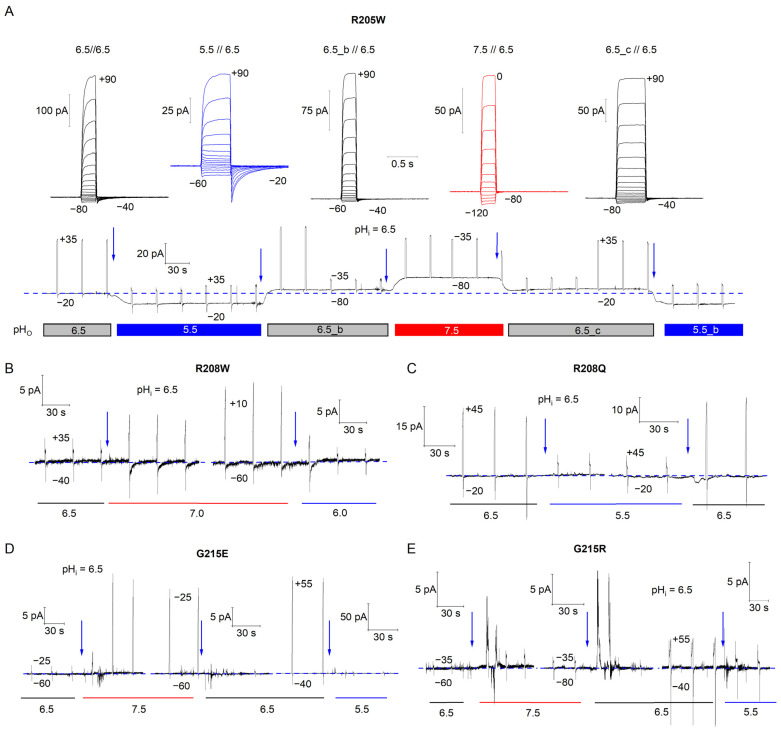
The R205W mutant leaks protons in the closed state. Whole-cell recordings of test pulses during external pH (pH_o_) exchanges for the different mutants were performed. Cells were held at a potential where the channel is closed (*V*_hold_), the most negative voltage shown in each recording, and depolarized for 1 s to the most positive potential displayed (*V*_test_). To achieve steady-state recordings, the interval between each *V*_test_ was 30 s. The current at the holding potential (*I*_hold_) was monitored during the experiments. A change in *I*_hold_ (blue dashed line) indicates protons leaking through the channel in the closed state. Blue arrows indicate the moment when the pH_o_ change was applied. In all cases, the pipette solution pH (pH_i_) was 6.5. (**A**) Top: Families of depolarizing pulses for the same cell expressing the R205W mutant at different pH_o_//pH_i_ conditions. Pulses were applied in 10 mV increments from the most negative potential to the most positive potential shown. *V*_hold_ was −40 mV at symmetrical pH_o_ = pH_i_ = 6.5 (gray traces), −20 mV at pH_o_ 5.5 (blue traces), and −80 mV at pH_o_ 7.5 (red traces). Bottom: Recording of test pulses applied during pH_o_ exchanges from 6.5 (gray) to 5.5 (blue) and 7.5 (red). After every pH_o_ exchange, *I*_hold_ follows the direction of the proton driving force, indicating a proton leak in the closed state. In the other mutants, *I*_hold_ remains constant during the pH_o_ exchanges like those shown here, demonstrating that the R208W (**B**), R208Q (**C**), G215E (**D**), and G215R (**E**) mutants do not detectably conduct protons in the closed state.

**Figure 5 biomolecules-15-00156-f005:**
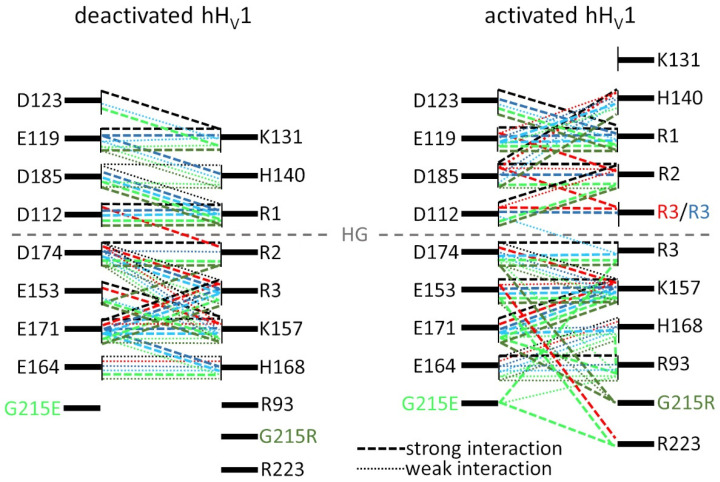
Salt bridges in the hH_V_1 channels. Salt bridges in the deactivated (left panel) and activated (right panel) channels are shown as dashed lines for strong interactions and as dotted lines for weaker interactions. Black, red, dark blue, light blue, light green, and dark green lines are for the WT, R205W, R208Q, R208W, G215E, and G215R channels, respectively. R3/R3 indicates the third voltage-sensing arginine of the R205W and R208Q mutants, which is located above the HG in the activated configuration.

**Figure 6 biomolecules-15-00156-f006:**
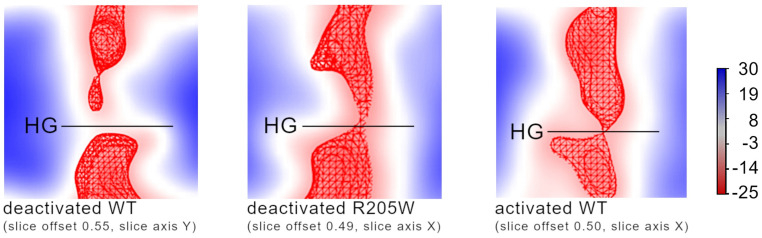
Electrostatic potential. The electrostatic potential maps and isosurfaces (depicted as wireframe) at an isovalue of the potential of −160 mV are shown in the region of the hydrophobic gasket for deactivated WT, deactivated R205W, and activated WT, from left to right. For more visibility, the wireframes are not colored according to their isovalue (−160 mV) but in red. The slice offsets and axes were chosen in order to show the plane for each channel that passes through the maximum number of side chains pointing towards the interior of the pore. The color scale on the right is in units of kT/e. At T = 310 K, one unit of electrostatic potential is equivalent to 27 mV and −160 mV corresponds to −6 kT/e. The interior of the channel is attractive for protons (red), except at the HG of deactivated WT, where the electrostatic potential is neutral (white).

**Figure 7 biomolecules-15-00156-f007:**
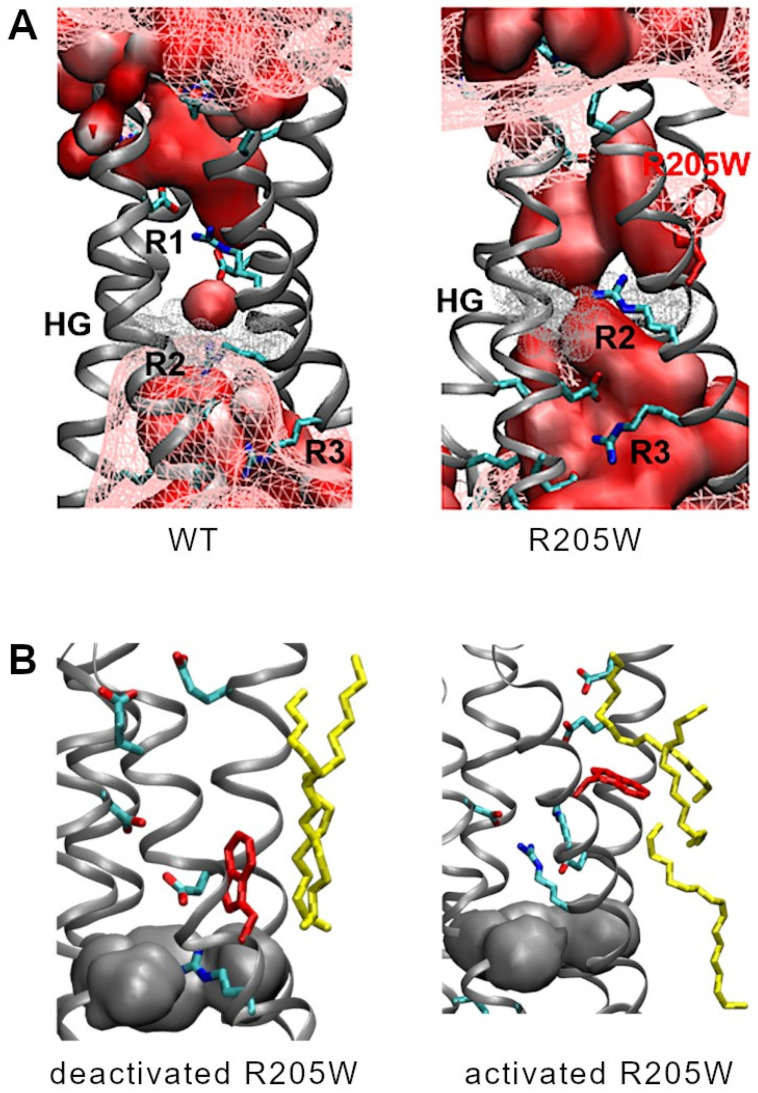
Electrostatic potential and water wire in the deactivated WT and R205W channels and (**B**) interaction of W205 with lipid. (**A**) The electrostatic potential at isovalue −160 mV (wireframe) and water wire (red surface) in deactivated WT (left) and R205W (right) are shown, as examples, in the centroid structure of cluster c0 for WT and c1 for R205W. In deactivated WT, the electrostatic potential and water wire are both discontinuous, almost in the whole external vestibule for the electrostatic potential and at R1 and R2, in a region comprising the HG (shown as a gray dot surface), for the water. In deactivated R205W, the electrostatic potential is generally continuous across the HG, as shown previously in Figure 6. In most structures of R205W, both the electrostatic potential and the water wire are simultaneously continuous across the HG. In a few structures where the electrostatic potential is not continuous, the water wire is continuous at the HG, as shown here for the centroid structure of the cluster. (**B**) Interactions between the mutated first voltage-sensing arginine R205 to tryptophan and the lipid in the deactivated and activation configurations.

**Figure 8 biomolecules-15-00156-f008:**
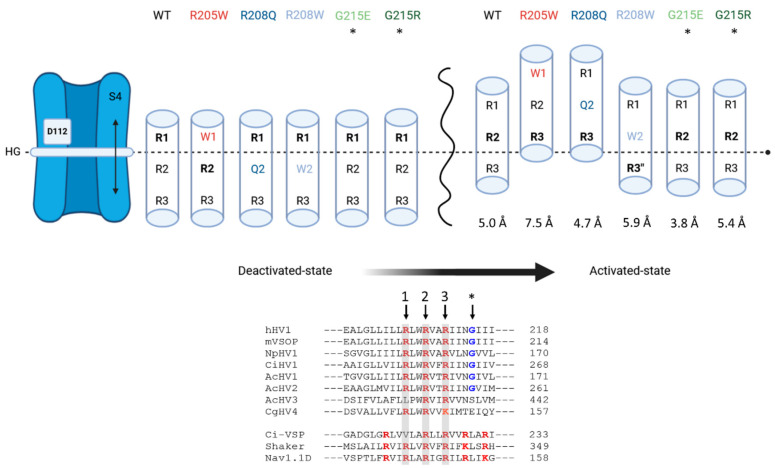
Positioning of the gating charge carriers in S4 in the deactivated and activated states relative to the hydrophobic gasket based on MD simulations. Top: Lateral-view cartoon depicting the monomeric form of hH_V_1 is shown on the left side. The hydrophobic gasket (HG) is formed by a ring of hydrophobic residues located approximately at the center of the channel (dashed line). The position of the three positively charged arginines in S4 alpha helix, Arg205 (R1), Arg208 (R2), and Arg211 (R3), relative to the HG is shown. During hH_V_1 gating, the S4 backbone together with these charges move extracellularly from a deactivated state (left) to an activated state (right). Mutants with substitutions of any of the three gating charges are labeled accordingly. For both states, deactivated and activated, the residues presenting strong electrostatic interaction with Asp112 are bold highlighted or marked with “″” when the interaction is weak. The total S4 backbone shift from deactivated to activated configurations for each mutant is shown. Bottom: S4 multiple-sequence alignment of several hH_V_1 and related molecules (see Section 2). Amino acids at position 215 (or equivalent) are marked with “*” in both, Top and Bottom panels.

**Table 1 biomolecules-15-00156-t001:** Salt bridges in the hH_V_1 channels. Salt bridges in deactivated (d) and activated (a) channels are marked “**✓**” and “∨” for strong and weak interactions, respectively. No sign denotes no, or no significant, salt bridge. R1 is the voltage-sensing arginine at sequence position 205 (R205) and is a tryptophan residue in the R205W mutant channel; R2 is the arginine at sequence position 208 (R208) and is a glutamine or a tryptophan residue in the R208Q and R208W mutant channels, respectively.

	WT		R205W		R208Q		R208W		G215E		G215R	
	d	a	d	a	d	a	d	a	d	a	d	a
R1-D112	✓				✓		✓		✓		✓	
R1-E119		✓				✓		✓		✓		✓
R1-D123		✓				✓		∨		✓		✓
R1-D185	∨				✓		✓		✓		✓	
R2-D112		✓	✓	∨						✓		✓
R2-E119				✓								
R2-E153											✓	
R2-D174	✓				∨				✓		✓	
R2-D185		✓		∨		✓				✓		∨
R3-D112				✓		✓		∨				
R3-D174	∨	✓	✓		✓		✓	✓	∨	✓	∨	✓
R3-E171	✓		✓		✓		✓		✓		✓	
R3-D185				✓								
H140-E119				∨	✓	∨	∨	✓	∨	∨		
H140-D185	∨	✓		∨		∨						
H168-R93										✓		
H168-E164	∨	∨			∨	∨	∨	∨	✓	∨	∨	∨
H168-E171					✓	∨				∨		
R93-E164		✓				∨		∨		∨		∨
R93-E171										✓		
R93-D174										∨		
R223-E153				✓						✓		
K131-E119	✓				∨		∨		∨		✓	
K131-D123	✓						∨		✓			
K157-E153	✓	✓	✓	∨		✓	∨	✓	✓	✓	✓	✓
K157-E171	✓	✓	∨	✓	✓	✓	✓	✓	∨	✓	∨	✓
K157-D174	∨	∨	∨	✓	∨	∨	∨	∨	∨	∨	∨	∨
G215E-R93										∨		
G215E-R211										✓		
G215E-R223										✓		
G215R-E153												∨
G215R-E171												✓
G215R-D174												✓
Sum	7✓ 5∨	9✓ 2∨	4✓ 2∨	6✓ 5∨	7✓ 4∨	6✓ 6∨	5✓ 6∨	5✓ 5∨	7✓ 5∨	12✓ 7∨	7✓ 4∨	8✓ 5∨

## Data Availability

The original contributions presented in this study are included in the article. Further inquiries can be directed to the corresponding author.

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
