# Peer review of "Biophysical Properties of Somatic Cancer Mutations in the S4 Transmembrane Segment of the Human Voltage-Gated Proton Channel hHV1"

_biomolecules, 2025, doi:10.3390/biom15020156_

Round 1

Reviewer 1 Report

Comments and Suggestions for Authors

The authors carried out electrophysiological and molecular dynamic simulations to investigate the biophysical properties of the somatic mutations from Hv1 channel S4 transmembrane segment. The manuscript is well-written, logically structured, and clearly presented.

Minor comments:

1.      Figure 2A: The representative traces for the wild-type (WT) and several mutants should be displayed with consistent size dimensions. For example, the trace for G215E appears significantly larger than the other traces. To ensure clarity and uniformity, the authors are advised to adjust the size of all six traces to be visually consistent with one another.

2.      Figure 2B-2E:  The information conveyed by these four graphs is less critical than that presented in Figure 2A. Therefore, it is recommended that the sizes of Figures 2B-2E be reduced. Alternatively, given that Figure 2A provides substantial and important biophysical information about the WT and mutant Hv1 channels, the authors should consider allocating more space to Figure 2A to better display its details.

3.      Line 529-530:  According to the supplementary data (Figure S2MD RMSF graph), it appears that G215R exhibits the largest deviation relative to the WT, rather than G215E. Please review and verify this observation to ensure the accuracy of their interpretation.

Author Response

  1. Figure 2A: The representative traces for the wild-type (WT) and several mutants should be displayed with consistent size dimensions. For example, the trace for G215E appears significantly larger than the other traces. To ensure clarity and uniformity, the authors are advised to adjust the size of all six traces to be visually consistent with one another.

We thank the referee for the valuable insight. Following the recommendation, we have modified Fig. 2. The families of currents in Fig. 2A are now adjusted to similar sizes, allowing for better visualization of details. The traces for each mutant now appear more consistent with one another.

  1. Figure 2B-2E: The information conveyed by these four graphs is less critical than that presented in Figure 2A. Therefore, it is recommended that the sizes of Figures 2B-2E be reduced. Alternatively, given that Figure 2A provides substantial and important biophysical information about the WT and mutant Hv1 channels, the authors should consider allocating more space to Figure 2A to better display its details.

Thank you for your observation; we found it very helpful. In response, we have revised Fig. 2 entirely. Along with making the traces in panel A more uniform and larger, we reduced the size of the graphs in panels B through E. These adjustments have significantly improved the figure's overall appearance and readability.

  1. Line 529-530: According to the supplementary data (Figure S2MD RMSF graph), it appears that G215R exhibits the largest deviation relative to the WT, rather than G215E. Please review and verify this observation to ensure the accuracy of their interpretation.

The referee is correct, and we apologize for the typographical error. We have corrected lines 529 and 530 accordingly. The revised text now reads: "Only one larger exception is observed for deactivated G215R. Also larger (but smaller than for G215R) deviations...". Thank you for the observation.

Reviewer 2 Report

Comments and Suggestions for Authors

The paper " Biophysical properties of somatic mutations in the S4 transmembrane segment of the human voltage-gated proton channel hHV1, and their possible implications in cancer progression" by Derst et al describes aims to analyze several mutations in a proton channel and to relate them to cancer progression. They found two primary hotspots for somatic mutations within the hHV1 channel: the central N-terminal region and the S4 transmembrane domain, which includes the voltage-sensor motif. They further characterized five specific mutations within the S4 segment. These mutations were found to alter channel function, with R205W and R208W showing significant changes in gating while maintaining proton selectivity. In addition, all tested mutants exhibited reduced effective charge displacement and proton conduction. The authors concluded that such mutations may affect the efficacy of hHV1 in extruding protons, which is critical for maintaining pH homeostasis in cancer cells. Overall, the findings indicate that somatic mutations in the hHV1 channel can significantly affect its function and may have important implications for cancer biology and treatment strategies. Overall, the study is of interest and is well conducted. There are some concerns that should be addressed:

Major points:

1.       There is no experimental evidence that connects the mutations to cancer. Do cancer cells show altered physiological behavior with these mutations? The whole framing of the work should be focused on the biophysical analysis of mutations. This includes modifying the title accordingly.

2.       In the context – what is the frequency of these mutations in non-cancer cells?

3.       Only S4 mutations were experimentally tested. The justification for this approach is not clear, as the role of S4 is well established. I would suggest adding analysis of mutations from the other hotspot as well.

4.       Throughout the paper there is no proper statistical analysis that will help evaluating the data in respect to mutations effects on various parameters.

5.       MD simulations – how the membrane potential was considered in the simulations?

6.       The paper lacks a discussion of the underlying mechanisms by which these mutations influence channel activity and even more, how they may affect cancer pH homeostasis. A more thorough mechanistic understanding could strengthen the conclusions drawn.

Minor points:

7.       Fig. 2A, please use consistent time scale to enable comparison. Also, some collected data and statistics are necessary to evaluate the data.

8.       Evaluation of gating charge is better done by measuring gating currents. Have the authors tried that?

9.       Fig. 3. Please provide individual values rather than bars (scatter plots with mean) and statistics for evaluating significance.

10.   Point 4.4 in the discussion is not clear. I am not aware of a claim that the voltage dependence can by attributed to a simple arginine count

Author Response

Major points:

  1. There is no experimental evidence that connects the mutations to cancer.

We appreciate the reviewer’s comment. For sure, we have no access to patient material including the respective tumor cells. Nevertheless, all the mutations analyzed in this manuscript are documented as occurring in cancer cells, supporting their connection to cancer. Please note Supplemental Table S3. We have also discussed our view that the publicly documented mutations in hHV1 are unlikely to be driver mutations. Instead, they are more likely passenger mutations, potentially deleterious ones. This conclusion is further supported by our biophysical analysis of the mutated channel.

Do cancer cells show altered physiological behavior with these mutations?

We thank the reviewer for this comment. As noted earlier, the mutations identified are present in human cancer cells, as shown in Supplemental Table S3. While we did not analyze the extent to which the mutated channel affects the proliferation of the cells used in this study, we recognize the importance of this question and plan to conduct such studies in the future to evaluate the impact on highly proliferative cells. Given the already extensive scope of this study, which focuses on the biophysical properties, we agree that these experiments would be valuable in future work.

The whole framing of the work should be focused on the biophysical analysis of mutations. This includes modifying the title accordingly.

We thank the reviewer for this insightful comment. The reviewer is correct that the majority of our data focuses on the biophysical and structural analysis of the channel. This analysis is highly detailed, with the model accurately reproducing the electrophysiological data. Furthermore, we were able to calculate perturbation energies and explain the changes induced by amino acid substitutions. Importantly, the mutations analyzed in our study are those documented in publicly accessible cancer databases, that specifically catalog mutations in voltage-gated proton channels. This suggestive connection to cancer is supported by the literature, as somatic mutations, has been extensively studied and our data align well with existing findings. We agree with the reviewer that despite the existence of a large body of suggestive work, the mechanisms by which HV channels are involved in the progression of cancer remain to be determined.

With this in mind, we have carefully worded the title: “Biophysical properties of somatic mutations in the S4 transmembrane segment of the human voltage-gated proton channel hHV1, and their possible implications in cancer progression.” Our study closely examines the biophysical consequences of a number of mutations that have been identified in cancer patients, with the intention that this knowledge will contribute to the future elucidation of the mechanisms involved. By clearly including the word “possible” we show that we realize that this connection remains to be tested directly, but if we do not mention the idea that much existing evidence supports involvement of HV channels in several types of cancer, the point of the study would not be clear.

  1. In the context – what is the frequency of these mutations in non-cancer cells?

We thank the reviewer for this important comment. Indeed, we do not currently know the frequency of mutations in the voltage-gated proton channel in non-cancer cells. To the best of our knowledge, the mutation rate is approximately 0.003 per genome per cell generation, which is ten times lower than the frequency reported in Supplemental Figure 3. However, we would not dismiss the possibility that a previously healthy cell could, by chance, develop such a mutation.

  1. Only S4 mutations were experimentally tested. The justification for this approach is not clear, as the role of S4 is well established. I would suggest adding analysis of mutations from the other hotspot as well.

We sincerely thank the reviewer for their insightful comment. We share the enthusiasm for thoroughly investigating all the mutations we identified. However, given the extensive amount of data already included in this manuscript, we opted to focus our efforts on the S4 region for this study. To our knowledge, the role of the S4 segment, particularly in the voltage-gated proton channel, is not yet fully established. Numerous studies have explored the S4's involvement in channel gating, often yielding partially contrasting findings ,  (Ramsey et al., 2006; Sasaki et al., 2006; Ramsey et al., 2010; Fujiwara et al., 2013; Gonzalez et al., 2013; Chamberlin et al., 2014; Kurokawa and Okamura, 2014; Takeshita et al., 2014; Chamberlin et al., 2015; Cherny et al., 2015; Okamura et al., 2015; Okuda et al., 2016; Randolph et al., 2016; van Keulen et al., 2017; Carmona et al., 2018; De La Rosa and Ramsey, 2018; Bayrhuber et al., 2019; Geragotelis et al., 2020; Carmona et al., 2021; Jardin et al., 2022; Chaves et al., 2023), to name just a few.

In our study, we believe we have contributed valuable insights into this region. For example, we observed that the proton leak conductance uniquely occurs when the first arginine (R205) is substituted. Interestingly, another arginine (R208) does not produce a proton leak, even with two contrasting substitutions. This is a novel finding that has not been previously reported. Additionally, we discovered that substitutions distant from the arginines could alter the gating charge, an unexpected result. Furthermore, the R208W substitution significantly slows the overall gating kinetics, suggesting an interaction between tryptophan and the hydrophobic gasket (Banh et al., 2019).

Importantly, all the S4 mutations we investigated are pathophysiologically relevant, as they have been identified in real patients suffering from cancer. We fully agree with the reviewer’s suggestion to analyze the other hotspot mutations, and we consider this a key objective for future research. Pending availability of manpower and funding, we also aim to systematically study all somatic mutations documented in the databases. We appreciate the reviewer’s encouragement and align with their perspective that the second hotspot is a particularly compelling avenue for further.

  1. Throughout the paper there is no proper statistical analysis that will help evaluating the data in respect to mutations effects on various parameters.

We appreciate the referee's observation. We generally avoid the excessive or unnecessary use of statistical significance tests, which are often misinterpreted by authors and readers, as thoroughly discussed in the literature (Colquhoun, 2017; Selke et al., 2001). Instead, we ensure that all data throughout the paper are presented as mean ± S.D./S.E.M., with the population size referenced in each figure. However, in line with the referee's recommendation, we have conducted statistical significance tests for each mutant compared to the WT hHV1 channel. These analyses were applied to the conduction/gating parameters shown in Fig. 2, as well as the kinetic behaviours displayed in Fig. 3. Both figures have been revised accordingly, and a note on the p-values has been added to the description of each figure.

  1. MD simulations – how the membrane potential was considered in the simulations?

There is no voltage potential across the membrane in all our simulations.

Simulating a potential across the membrane would (or at least should) favor the population of more closed or open conformations, depending on the negative or positive sign of the potential across the membrane. This approach was used for example in MD simulations of the homologous channel CiVSP, that has the highest sequence homology to hHv1 in the transmembrane domain, to bias the free energy landscape toward a down-minus and toward an up-plus state (Shen et al., 2022), or to simulate the voltage-dependent conformational change from the hyperpolarized to the depolarized conformations of hHv1 (Geragotelis et al., 2020). Here, we were interested in characterizing a deactivated and activated states that are in agreement with the conformations that we identified in our simulations of the (∆)pH-dependent gating of hHv1. As in e.g. (Shen et al., 2022), such states are properly identified in MD simulations without potential across the membrane.

  1. The paper lacks a discussion of the underlying mechanisms by which these mutations influence channel activity and even more, how they may affect cancer pH homeostasis. A more thorough mechanistic understanding could strengthen the conclusions drawn.

We are thankful for the referee's comment. The discussion section of the manuscript spans five pages, focusing extensively on the biophysical behavior of the different channels compared to the WT channel and linking these observations to the molecular behaviors identified in our MD simulations. In this study, we measured and calculated gating, conductive, and kinetic parameters for each analyzed mutant. Our MD simulations provided insights into interactions between substituted residues and other structural components of the channel, reinforcing the electrophysiological findings.

Additionally, we implemented a strategy to calculate the perturbation energies resulting from single amino acid substitutions. These energetic parameters, combined with molecular dynamics, allowed us to compare our findings with the energies of different electrostatic interactions reported in the literature. For instance, this approach enabled us to evaluate potential van der Waals interactions between the chemical groups of Trp mutants and other critical structural components of the channel (see section 4.2).

The comprehensive electrophysiological characterization presented in this study is sufficient to conclude that the mutations analyzed, relative to WT hHV1, result in defective proton channels. The mutants exhibit lower H⁺ conductance per unit of membrane area (Fig. 2B), reduced charge displacement under the influence of the electrical field (Fig. 2C), and, in some cases, activation thresholds (Vthres) are more negative than the reversal potential (Vrev), enabling proton inward currents (Fig. 2D).

The kinetic analysis provides further insights. Except for R205W, the energetic patterns of the mutants indicate an increased energetic barrier for the close-to-open transition, favouring the closed state (Fig. 3D). Additionally, our measurements reveal that one of the channels, R205W, leaks protons in the closed state. Supported by our MD simulations, we propose a molecular explanation for this H⁺ leakage, involving a continuous proton-attractive potential that spans the hydrophobic gasket (Figs. 6 and 7). This is discussed in detail in section 4.3.

The observed effects compromise the typical function of HV1, which includes proton extrusion (cytosolic pH homeostasis) and charge compensation (DeCoursey, 2003). While our study did not include proliferation assays or in-vivo analysis of cytosolic pH regulation in tumor cells expressing the HV1 mutants, there is substantial evidence emphasizing the critical role of HV1 channels in acid/proton extrusion for tumor survival and progression (Wang et al., 2011; Wang et al., 2012; Hondares et al., 2014; Asuaje et al., 2017). Nonetheless, as the referee rightly suggested, further in vivo studies would be valuable to draw more definitive conclusions regarding the specific effects of each mutation on tumor development. We sincerely appreciate the referee’s insightful observation.

Minor points:

  1. Fig. 2A, please use consistent time scale to enable comparison. Also, some collected data and statistics are necessary to evaluate the data.

We appreciate the referee's suggestion. Initially, we considered displaying all representative records with uniform time and current scale bars to favour comparison. However, due to the significant differences in kinetics and conduction among the channels, we decided to present each family of pulses for each mutant using individualized scale bars. This approach ensures that readers can clearly observe the details of the proton currents; especially important for those proton channel enthusiasts. If we had used the same scale bars throughout, Fig. 2 would appear as follows:

Accordingly, we revised Fig. 2A as recommended by Referee #1, as we agree that this adjustment allows for better focus on the details. Nevertheless, following your advice, we included a new supplementary figure (Fig. S2) that presents the same records from Fig. 2A with a common scale bar to enable comparison. A brief note regarding this change has been added to the description of Fig. 2A. Please refer to the supplementary information for details. Additionally, we implemented a statistical significance test on the parameters shown in Fig2B-E and added a comment on this regard in the figure’s description.

  1. Evaluation of gating charge is better done by measuring gating currents. Have the authors tried that?

In principle, gating charge movement can be determined by measuring gating currents. Gating current measurements have been reported in proton channels by using a non-conductive mutant in sea squirt CiHV1 (Carmona et al., 2018) and in humans (De La Rosa and Ramsey, 2018). However, these measurements are uniquely problematic for HV channels for several reasons, as discussed in a critique of these two papers (DeCoursey, 2018). Although it appears that both studies reported genuine gating currents, due to the small unitary conductance of HV channels, neither group was able to determine the gating charge per channel (which requires determining the number of channels present). So in fact to date, no one has been able to use gating currents to determine the charge moved by a single HV channel.

  1. Fig. 3. Please provide individual values rather than bars (scatter plots with mean) and statistics for evaluating significance.

We thank the referee for the recommendation. In response, we have updated graphs B and C to scatter plots with mean values and have added a significance test. However, we did not modify the bar graphs in D and E, as they allow for a clear comparison of the activation (solid bars) and deactivation (open bars) parameters for the same mutant. We appreciate the referee's suggestion, which has improved the overall the presentation and readability of Fig. 3.

  1. Point 4.4 in the discussion is not clear. I am not aware of a claim that the voltage dependence can by attributed to a simple arginine count

 We thank the reviewer for this comment. We have revised the title of this paragraph to avoid potential misinterpretation. Our intention was to highlight that charges located away from the inner core of the channel, and likely beyond the focus of the electric field, can still influence the gating charges determined by the limiting slope technique.

The connection between the gating charge changes observed with the somatic mutations G215E and G215R remains unclear. We hypothesize that alterations in the salt-bridge network may be related to the observed changes in gating. An outward displacement of S4 moves charges in one direction, thereby generating a current. If numerous positive charges are concentrated on S4 and are able to cross the focus of the electric field, the resulting current—and consequently, the calculated gating charge—would increase.

Our experimental data, however, contradicts this simplified view. Despite our efforts, we were unable to conclusively resolve this issue (lines 988–991). We appreciate the reviewer’s suggestion and hope the revised title is now clearer and less misleading.

References

Asuaje et al., “The inhibition of voltage-gated H+ channel (HVCN1) induces acidification of leukemic Jurkat T cells promoting cell death by apoptosis,” Pflugers Arch - Eur J Physiol, vol. 469, no. 2, pp. 251–261, Feb. 2017, doi: 10.1007/s00424-016-1928-0.

Bayrhuber, M., I. Maslennikov, W. Kwiatkowski, A. Sobol, C. Wierschem, C. Eichmann, L. Frey, and R. Riek. 2019. Nuclear Magnetic Resonance Solution Structure and Functional Behavior of the Human Proton Channel. Biochemistry. 58:4017-4027. doi:10.1021/acs.biochem.9b00471

Carmona, E.M., M. Fernandez, J.J. Alvear-Arias, A. Neely, H.P. Larsson, O. Alvarez, J.A. Garate, R. Latorre, and C. Gonzalez. 2021. The voltage sensor is responsible for DeltapH dependence in H(v)1 channels. Proc Natl Acad Sci U S A. 118:e2025556118. doi:10.1073/pnas.2025556118

Carmona, E.M., H.P. Larsson, A. Neely, O. Alvarez, R. Latorre, and C. Gonzalez. 2018. Gating charge displacement in a monomeric voltage-gated proton (H(v)1) channel. Proc Natl Acad Sci U S A. 115:9240-9245. doi:10.1073/pnas.1809705115

Chamberlin, A., F. Qiu, S. Rebolledo, Y. Wang, S.Y. Noskov, and H.P. Larsson. 2014. Hydrophobic plug functions as a gate in voltage-gated proton channels. Proc Natl Acad Sci U S A. 111:E273-282. doi:10.1073/pnas.1318018111

Chamberlin, A., F. Qiu, Y. Wang, S.Y. Noskov, and H.P. Larsson. 2015. Mapping the gating and permeation pathways in the voltage-gated proton channel Hv1. J Mol Biol. 427:131-145. doi:10.1016/j.jmb.2014.11.018

Chaves, S. Bungert‐Plümke, A. Franzen, I. Mahorivska, and B. Musset, “Zinc modulation of proton currents in a new voltage-gated proton channel suggests a mechanism of inhibition,” The FEBS Journal, vol. 287, no. 22, pp. 4996–5018, 2020, doi: https://doi.org/10.1111/febs.15291.

Chaves, G., A.G. Ayuyan, V.V. Cherny, D. Morgan, A. Franzen, L. Fieber, L. Nausch, C. Derst, I. Mahorivska, C. Jardin, T.E. DeCoursey, and B. Musset. 2023. Unexpected expansion of the voltage-gated proton channel family. FEBS J. 290:1008-1026. doi:10.1111/febs.16617

Cherny, V.V., D. Morgan, B. Musset, G. Chaves, S.M. Smith, and T.E. DeCoursey. 2015. Tryptophan 207 is crucial to the unique properties of the human voltage-gated proton channel, hHV1. J Gen Physiol. 146:343-356. doi:10.1085/jgp.201511456.

Colquhoun D. The reproducibility of research and the misinterpretation of p-values. R Soc Open Sci. 2017 Dec 6;4(12):171085. doi: 10.1098/rsos.171085. Erratum in: R Soc Open Sci. 2018 Mar 7;5(3):180100. doi: 10.1098/rsos.180100. PMID: 29308247; PMCID: PMC5750014.

DeCoursey TE. Gating currents indicate complex gating of voltage-gated proton channels. Proc Natl Acad Sci U S A. 2018 Sep 11;115(37):9057-9059. doi: 10.1073/pnas.1813013115. Epub 2018 Aug 22. PMID: 30135099; PMCID: PMC6140469.

Decoursey TE. Voltage-gated proton channels and other proton transfer pathways. Physiol Rev. 2003 Apr;83(2):475-579. doi: 10.1152/physrev.00028.2002. Erratum in: Physiol Rev. 2003 Jul;83(3):1067. Erratum in: Physiol Rev. 2004 Oct;84(4):1479. PMID: 12663866.

De La Rosa, V., and I.S. Ramsey. 2018. Gating Currents in the Hv1 Proton Channel. Biophys J. 114:2844-2854. doi:10.1016/j.bpj.2018.04.049

Fujiwara et al., “The cytoplasmic coiled-coil mediates cooperative gating temperature sensitivity in the voltage-gated H(+) channel Hv1,” Nat Commun, vol. 3, p. 816, May 2012, doi: 10.1038/ncomms1823.

Fujiwara, Y., T. Kurokawa, K. Takeshita, A. Nakagawa, H.P. Larsson, and Y. Okamura. 2013. Gating of the designed trimeric/tetrameric voltage-gated H+ channel. J Physiol. 591:627-640. doi:10.1113/jphysiol.2012.243006

Geragotelis, A.D., M.L. Wood, H. Goddeke, L. Hong, P.D. Webster, E.K. Wong, J.A. Freites, F. Tombola, and D.J. Tobias. 2020. Voltage-dependent structural models of the human Hv1 proton channel from long-timescale molecular dynamics simulations. Proc Natl Acad Sci U S A. 117:13490-13498. doi:10.1073/pnas.1920943117

Gonzalez, C., S. Rebolledo, M.E. Perez, and H.P. Larsson. 2013. Molecular mechanism of voltage sensing in voltage-gated proton channels. J Gen Physiol. 141:275-285. doi:10.1085/jgp.201210857

Hondares et al., “Enhanced activation of an amino-terminally truncated isoform of the voltage-gated proton channel HVCN1 enriched in malignant B cells,” Proc. Natl. Acad. Sci. U.S.A., vol. 111, no. 50, pp. 18078–18083, Dec. 2014, doi: 10.1073/pnas.1411390111.

Jardin, C., N. Ohlwein, A. Franzen, G. Chaves, and B. Musset. 2022. The pH-dependent gating of the human voltage-gated proton channel from computational simulations. Phys Chem Chem Phys. 24:9964-9977. doi:10.1039/d1cp05609c

Kurokawa, T., and Y. Okamura. 2014. Mapping of sites facing aqueous environment of voltage-gated proton channel at resting state: a study with PEGylation protection. Biochim Biophys Acta. 1838:382-387. doi:10.1016/j.bbamem.2013.10.001

Okamura, Y., Y. Fujiwara, and S. Sakata. 2015. Gating mechanisms of voltage-gated proton channels. Annual review of biochemistry. 84:685-709. doi:10.1146/annurev-biochem-060614-034307

Okuda, H., Y. Yonezawa, Y. Takano, Y. Okamura, and Y. Fujiwara. 2016. Direct Interaction between the Voltage Sensors Produces Cooperative Sustained Deactivation in Voltage-gated H+ Channel Dimers. J Biol Chem. 291:5935-5947. doi:10.1074/jbc.M115.666834

Ramsey, I.S., Y. Mokrab, I. Carvacho, Z.A. Sands, M.S.P. Sansom, and D.E. Clapham. 2010. An aqueous H+ permeation pathway in the voltage-gated proton channel Hv1. Nat Struct Mol Biol. 17:869-875. doi:10.1038/nsmb.1826

Ramsey, I.S., M.M. Moran, J.A. Chong, and D.E. Clapham. 2006. A voltage-gated proton-selective channel lacking the pore domain. Nature. 440:1213-1216. doi:10.1038/nature04700

Randolph, A.L., Y. Mokrab, A.L. Bennett, M.S. Sansom, and I.S. Ramsey. 2016. Proton currents constrain structural models of voltage sensor activation. Elife. 5. doi:10.7554/eLife.18017

Sasaki, M., M. Takagi, and Y. Okamura. 2006. A voltage sensor-domain protein is a voltage-gated proton channel. Science. 312:589-592. doi:10.1126/science.1122352.

Sellke, T., Bayarri, M. J., & Berger, J. O. (2001). Calibration of ρ Values for Testing Precise Null Hypotheses. The American Statistician, 55(1), 62–71. https://doi.org/10.1198/000313001300339950

Shen R, Meng Y, Roux B and Perozo E. Mechanism of voltage gating in the voltage-sensing phosphatase Ci-VSP. Proc Natl Acad Sci USA 2022, 119: e2206649119. doi: 10.1073/pnas.2206649119

Takeshita, K., S. Sakata, E. Yamashita, Y. Fujiwara, A. Kawanabe, T. Kurokawa, Y. Okochi, M. Matsuda, H. Narita, Y. Okamura, and A. Nakagawa. 2014. X-ray crystal structure of voltage-gated proton channel. Nat Struct Mol Biol. 21:352-357. doi:10.1038/nsmb.2783

van Keulen, S.C., E. Gianti, V. Carnevale, M.L. Klein, U. Rothlisberger, and L. Delemotte. 2017. Does Proton Conduction in the Voltage-Gated H(+) Channel hHv1 Involve Grotthuss-Like Hopping via Acidic Residues? J Phys Chem B. 121:3340-3351. doi:10.1021/acs.jpcb.6b08339

Wang, S. J. Li, J. Pan, Y. Che, J. Yin, and Q. Zhao, “Specific expression of the human voltage-gated proton channel Hv1 in highly metastatic breast cancer cells, promotes tumor progression and metastasis,” Biochem. Biophys. Res. Commun., vol. 412, no. 2, pp. 353–359, Aug. 2011, doi: 10.1016/j.bbrc.2011.07.102.

Wang, S. J. Li, X. Wu, Y. Che, and Q. Li, “Clinicopathological and biological significance of human voltage-gated proton channel Hv1 protein overexpression in breast cancer,” J. Biol. Chem., vol. 287, no. 17, pp. 13877–13888, Apr. 2012, doi: 10.1074/jbc.M112.345280.

Round 2

Reviewer 2 Report

Comments and Suggestions for Authors

While the authors addressed the techniqual concerns I raised, I still feel that the connection to cancer is quite weak given tha data they provide. I would suggest changing the focus and framing of the study so it will not emphasize the connection to cancer.

Author Response

Open Review

Reviewer 2

While the authors addressed the techniqual concerns I raised, I still feel that the connection to cancer is quite weak given tha data they provide. I would suggest changing the focus and framing of the study so it will not emphasize the connection to cancer.

Submission Date

03 December 2024

Date of this review

25 Dec 2024 12:08:37

Response

We thank the reviewer for the recommendation. We agree with the referee's point that the scope of this study focuses on the biophysical properties of the selected mutants, as stated in the title. We also acknowledge that our data does not directly establish a connection between the characterized HV1 mutants and the progression of cancer, as we did not perform in vivo experiments to address this specific question.

The link to cancer is based on the somatic mutations selected for our study, which are documented in accessible cancer databases. Accordingly, we have been explicit in our wording, emphasizing that while the connection to cancer is plausible, it is not definitive. We have also made it clear that this study serves as a foundation for future research, as previous studies support the idea of a link between HV1 and cancer. This position is further substantiated in the following publications:

  1. Wang, Y., Zhang, S., & Li, S. J. (2013). Zn(2+) induces apoptosis in human highly metastatic SHG-44 glioma cells through inhibiting activity of the voltage-gated proton channel Hv1. Biochemical and Biophysical Research Communications, 438, 312–317. doi:10.1016/j.bbrc.2013.07.067
  2. Wang, Y., Wu, X., Li, Q., Zhang, S., & Li, S. J. (2013). Human voltage-gated proton channel Hv1: a new potential biomarker for diagnosis and prognosis of colorectal cancer. PLoS One, 8, e70550. doi:10.1371/journal.pone.0070550
  3. Wang, Y., Li, S. J., Wu, X., Che, Y., & Li, Q. (2012). Clinicopathological and biological significance of human voltage-gated proton channel Hv1 protein overexpression in breast cancer. Journal of Biological Chemistry, 287, 13877–13888. doi:10.1074/jbc.M112.345280
  4. Wang, Y., Li, S. J., Pan, J., Che, Y., Yin, J., & Zhao, Q. (2011). Specific expression of the human voltage-gated proton channel Hv1 in highly metastatic breast cancer cells promotes tumor progression and metastasis. Biochemical and Biophysical Research Communications, 412, 353–359. doi:10.1016/j.bbrc.2011.07.102
  5. Ventura, C., Leon, I. E., Asuaje, A., Martín, P., Enrique, N., Núñez, M., Cocca, C., & Milesi, V. (2020). Differential expression of the long and truncated Hv1 isoforms in breast cancer cells. Journal of Cellular Physiology, 235, 8757–8767. doi:10.1002/jcp.29719
  6. Fernandez, A., Pupo, A., Mena-Ulecia, K., & Gonzalez, C. (2016). Pharmacological Modulation of Proton Channel Hv1 in Cancer Therapy: Future Perspectives. Molecular Pharmacology, 90, 385–402. doi:10.1124/mol.116.103804

In summary, we acknowledge that the cancer connection is not definitive, we propose it as a promising lead for further exploration, supported by existing literature.

In response to the reviewer's recommendation, we have revised several sections of the manuscript to shift the focus away from cancer progression and emphasize the impairment of HV1 proton extrusion function. These changes include:

  • Title: We removed references to potential implications for cancer progression and rephrased the title to emphasize the study of the biophysical properties of cancer-related somatic mutants.
  • Lines 26–27, 83–84: We clarified the potential for impaired cellular homeostasis resulting from HV1 proton extrusion dysfunction.
  • Conclusion: The conclusion section has been completely rewritten to align with the reviewer's suggestion.
